# LayerCraft: Enhancing Text-to-Image Generation with CoT Reasoning and Layered Object Integration

**Yuyao ZHANG**
Dartmouth College

**Jinghao LI***
CUHK

**Yu-Wing TAI**
Dartmouth College

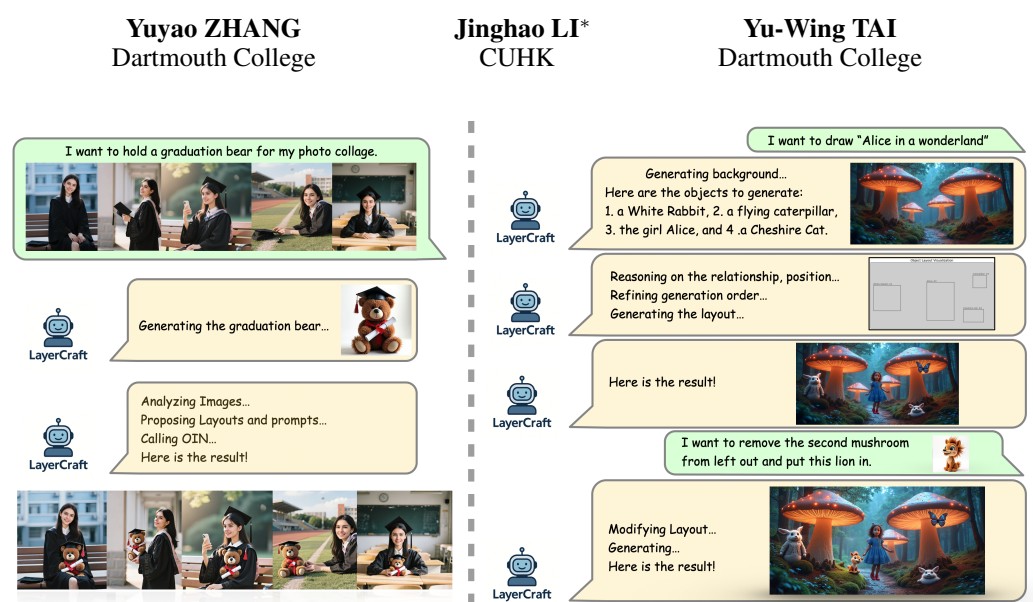

Figure 1: Application demonstrations for LayerCraft. **Left:** Demonstrates batch collage editing capabilities. A user uploads graduation photos and LayerCraft seamlessly integrates a graduation bear across all images. The system first generates a reference bear for consistency, then analyzes optimal placement while preserving facial identity and background integrity. **Right**: Illustrates the structured text-to-image generation process. From a simple "Alice in Wonderland" prompt, LayerCraft employs chain-of-thought reasoning to sequentially generate background elements, determine object layout, and compose the final image. The framework supports post-generation customization, as shown with the lion integration.

## Abstract

Text-to-image (T2I) generation has made remarkable progress, yet existing systems still lack intuitive control over spatial composition, object consistency, and multi-step editing. We present **LayerCraft**, a modular framework that uses large language models (LLMs) as autonomous agents to orchestrate structured, layered image generation and editing. LayerCraft supports two key capabilities: (1) *structured generation* from simple prompts via chain-of-thought (CoT) reasoning, enabling it to decompose scenes, reason about object placement, and guide composition in a controllable, interpretable manner; and (2) *layered object integration*, allowing users to insert and customize objects—such as characters or props—across diverse images or scenes while preserving identity, context, and style. The system comprises a coordinator agent, the **ChainArchitect** for CoT-driven layout planning, and the **Object Integration Network (OIN)** for seamless image editing using off-the-shelf T2I models without retraining. Through applications like batch collage editing and narrative scene generation, LayerCraft empowers non-experts to iteratively design, customize, and refine visual content with minimal manual effort. Code will be released at `https://github.com/PeterYYZhang/LayerCraft`.

---

*This work was done while Jinghao Li was an exchange student at Dartmouth College.

39th Conference on Neural Information Processing Systems (NeurIPS 2025).

# 1 Introduction

Text-to-image (T2I) generation has rapidly evolved with advances in diffusion models [14, 35, 32], transformer-based architectures [42], and scalable encoder-decoder frameworks [36]. Recent systems [9, 4, 6] produce visually impressive results from simple prompts. However, they still fall short in offering *precise, intuitive control* over spatial composition, multi-object interactions, and iterative customization.

Existing approaches to fine-grained T2I control often require architectural modifications or fine-tuning [55, 53, 37], which limits generality and usability. Others support instance-level manipulation [44, 48, 17], but often falter in complex scenes or suffer from spatial inconsistency. More structured methods like LayoutGPT [10] and GenArtist [45] attempt procedural generation, but neglect 3D spatial reasoning or rely on inefficient pipelines with excessive external tools. Even advanced multi-modal agents like GPT-4o[2] fail to maintain background consistency or facial identity over multiple editing iterations.

**LayerCraft** is our answer to these limitations: a fully automatic, modular framework for structured T2I generation and editing, designed to balance expressive control, compositional accuracy, and system efficiency. LayerCraft treats image synthesis as a step-by-step reasoning process, orchestrated by a team of specialized agents that handle prompt interpretation, spatial planning, and object integration. As shown in Figure 1, our framework supports applications such as batch collage editing with consistent object insertion, and narrative-driven image generation using structured reasoning and layout planning.

- **LayerCraft Coordinator** serves as the central interface, managing interactions between users and agents. It processes instructions, coordinates agent outputs, and integrates user feedback throughout the generation process.
- **ChainArchitect** performs *chain-of-thought (CoT)* reasoning to decompose prompts into structured layout plans. It first generates the background, then infers a spatial layout, represented as a dependency-aware 3D scene graph, to determine bounding boxes and relationships among objects. This planning phase supports complex multi-object scenes and facilitates layer-wise, editable image construction.
- **Object Integration Network (OIN)** uses the original FLUX [18] T2I model to seamlessly inpaint objects into specific regions. By applying dual LoRA adapters, OIN integrates both background and reference conditions while preserving generative quality. Its attention-mixing mechanism ensures that inserted objects align contextually and stylistically with the base image.

LayerCraft introduces several advantages over prior work: (1) it eliminates the need for model fine-tuning or external tools, making it accessible and lightweight; (2) it offers interpretable, spatially aware image construction via CoT-guided layout planning; and (3) it supports consistent object editing across single or multiple images without sacrificing visual quality. Compared to LayoutGPT [10] and GenArtist [45], which struggle with spatial coherence and integration complexity, LayerCraft provides a unified, agent-based framework capable of general-purpose generation and editing.

Our experiments demonstrate that LayerCraft excels in various creative workflows, from narrative scene composition to iterative and batch image editing, empowering both experts and non-experts to produce controllable, high-quality images with minimal effort.

# 2 Related Work

**Controllable Image Generation.** Text-to-image (T2I) generation has seen rapid progress, led by advances in diffusion models—from pixel-space methods like GLIDE [30] and Imagen [38] to more efficient latent-space frameworks such as Stable Diffusion [35] and Raphael [50]. Enhancements in multimodal alignment (e.g., DALLE-2 [33], Playground [20]) and architectural designs (e.g., Diffusion Transformers [31], PixArt [6], FLUX [18]) have substantially improved the quality and diversity of generated content. However, fine-grained and interpretable control remains challenging, especially in scenes with multiple objects or complex layouts. Personalization methods like DreamBooth [37] and Textual Inversion [11] support user-specific concepts but require task-specific

---

[2]`https://openai.com/index/gpt-4o-system-card/`

fine-tuning. Structured control approaches such as ControlNet [55] and GLIGEN [21] offer spatial conditioning via edge maps or boxes but rely on detailed inputs and lack high-level scene reasoning.

Recent techniques like Raphael [50] improve specialization through expert models at the cost of high computation. Lightweight alternatives like Attend [5] reduce overhead but struggle with compositional complexity. Autoregressive frameworks (e.g., LlamaGen [39], Show-O [49], Janus-Pro [8]) explore prompt-based synthesis via language models but often lack spatial structure. Meanwhile, emerging MLLMs such as GPT-4o and Gemini 2.0 Flash[3] show generative promise, though they require substantial resources and offer limited layout control.

In contrast, LayerCraft introduces a modular, agent-based framework for structured multi-object generation and editing with minimal user input. The *ChainArchitect* employs chain-of-thought (CoT) reasoning to produce interpretable, 3D-aware layouts, enabling precise spatial planning without manual annotations. The *Object Integration Network (OIN)* leverages dual-LoRA fine-tuning on a pre-trained diffusion model (FLUX) to enable parameter-efficient object integration with strong visual fidelity and contextual coherence. Unlike methods such as OminiControl [40], which emphasize attention-based spatial aligned and subject driven generation, LayerCraft supports broader workflows, including image-guided inpainting, iterative editing, and batch collage generation, within a unified, lightweight pipeline that generalizes effectively across diverse scenarios.

**Agent-Based Generation.** The rise of large language models (LLMs) has greatly advanced zero- and few-shot learning across diverse domains [1, 41]. With multimodal training [2, 23, 60], LLMs have evolved into powerful agents for reasoning and creative generation [51, 47, 25, 26, 27, 16]. Among these, LayoutGPT [10] uses LLMs to generate spatial layouts from text prompts. While effective for simple scenes, its reliance on static layout models limits its ability to handle complex prompts and spatial relationships, particularly due to the absence of multi-step reasoning. Other frameworks like GenArtist [45] and LLM Blueprints [12] follow a "generate-then-edit" paradigm, refining initial layouts or images through external editing modules. This often leads to stylistic drift and unstable outputs, due to fragmented control and lack of shared context across steps.

In contrast, LayerCraft offers an integrated multi-agent framework that unifies layout planning and refinement, and iterative object integration. The *LayerCraft Coordinator* orchestrates agent interactions and incorporates user feedback throughout the process. The *ChainArchitect* improves upon LayoutGPT by applying chain-of-thought (CoT) reasoning to generate structured, 3D-aware layouts, enabling compositional planning without external layout tools. The *Object Integration Network (OIN)* complements this with image-guided inpainting via dual-LoRA fine-tuning on a pre-trained model, supporting adaptive, context-aware generation while maintaining high visual fidelity. Unlike modular pipelines that rely on third-party components or model modifications, LayerCraft remains self-contained and parameter-efficient, offering a robust and consistent user experience.

**Chain of Thought Reasoning.** Chain-of-thought (CoT) prompting has proven effective in improving language model reasoning by decomposing complex tasks into intermediate steps [46, 57]. However, in multimodal settings, existing CoT approaches often rely on model finetuning over specialized datasets [29, 58], limiting their applicability in zero-shot or flexible generation scenarios.

LayerCraft takes a different approach by incorporating CoT reasoning without requiring additional fine-tuning. The *LayerCraft Coordinator* uses CoT to iteratively revise and enrich user prompts, while the *ChainArchitect* applies CoT-style decomposition to translate high-level instructions into structured, 3D-aware layouts. This allows LayerCraft to reason over complex spatial relationships and multi-object configurations in a fully zero-shot, training-free setting. By leveraging CoT within a modular agent framework, LayerCraft achieves interpretable, step-wise control in multimodal image generation, offering a robust alternative to methods that depend on task-specific finetuning or static layout templates.

## 3 Methodology

This section elaborates the detailed design of LayerCraft, overviewed in Figure 2. Leveraging GPT-4o as the central coordinator, LayerCraft enables self-monitoring, user-agent interaction, and

---

[3]https://console.cloud.google.com/vertex-ai/publishers/google/model-garden/gemini-2.0-flash-001?inv=1&invt=AbxbDg

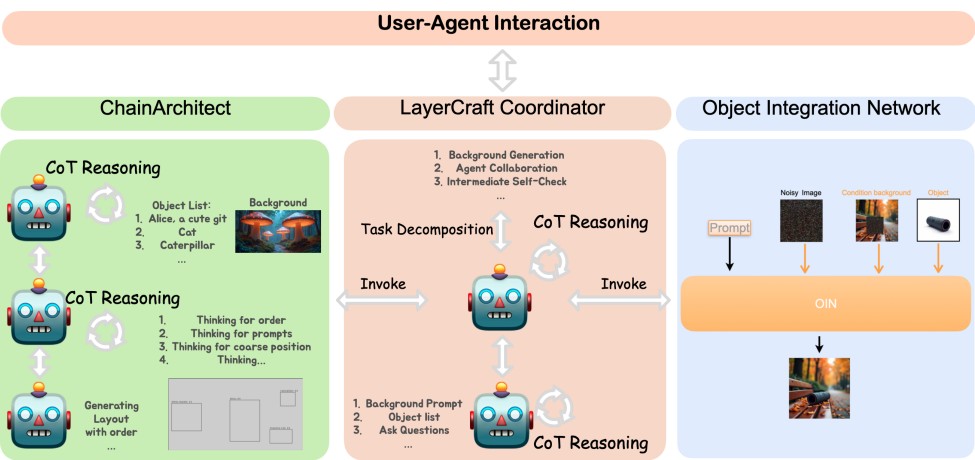

Figure 2: LayerCraft is a framework with three key components: the LayerCraft Coordinator, which processes user instructions and manages collaboration; ChainArchitect, which enhances prompts to plan layouts, identify objects and relationships, and assign bounding boxes using Chain-of-Thought reasoning; and the Object Integration Network (OIN), which enables image-guided inpainting for seamless object integration using the LoRA fine-tuned FLUX model.

aesthetically refined outputs and multi-turn editing. The framework consists of three main agents: (1) LayerCraft Coordinator, which processes user instructions and orchestrates agent collaboration; (2) ChainArchitect, a layout planning agent that generates backgrounds, assigns objects and their spatial relationships; and (3) Object Integration Network (OIN), which integrates objects seamlessly into the background based on given mask.

## 3.1 LayerCraft Coordinator

The LayerCraft Coordinator acts as the central orchestrator for the entire framework, overseeing the system's operation, ensuring smooth user-agent interactions, and directing agent collaboration. This component also serves as the primary interface for user input, streamlining communication between the user and the system.

**Agent-Agent Interaction** The framework integrates multiple specialized agents, each responsible for a specific task such as content recognition, reference image generation, layout planning (ChainArchitect), and final image generation and inpainting (OIN). The Coordinator plays a crucial role in orchestrating these agents, breaking down tasks, assigning responsibilities, and ensuring effective communication between them. Since generative models can produce intermediate outputs with inherent randomness, the Coordinator rigorously checks the consistency of both textual and visual outputs. If discrepancies are detected, it formulates corrective measures and delegates the task to the appropriate agent for regeneration. This enables LayerCraft to ensure that the final output meets the user's specifications.

**User-Agent Interaction** Although the system operates autonomously, users can modify or refine the output by interacting with the agents. For example, a user may request more details for a specific object or a customized layout for a particular region. The Coordinator facilitates multiple rounds of interaction, refining the image iteratively until the user's requirements are fully met.

**Chain-of-Thought (CoT) Enrichment.** The Coordinator enhances generation by iteratively enriching the text prompt using a Chain-of-Thought (CoT) approach. Starting from the user's input, it "asks itself" which objects should appear and how they should be arranged to meet user intent (see Figure 1, right). This reasoning produces detailed descriptions of background and foreground elements, filling gaps in the original prompt.

If the Coordinator determines the user's prompt is already sufficiently detailed, it skips CoT reasoning and proceeds directly to task delegation. This adaptive strategy improves efficiency by avoiding unnecessary steps when the input is complete.

## 3.2 ChainArchitect

ChainArchitect advances traditional layout generation models (e.g., LayoutGPT [10]) by integrating Chain-of-Thought (CoT) reasoning to better handle complex prompts involving multiple objects and intricate spatial relationships.

Given a user input prompt $P_i$, which may range from detailed to brief, the LLM identifies relevant objects and generates a structured list $O = \{O_i \mid i \in \mathbb{N}\}$ alongside a background description $P_{b_i}$. For instance, if the prompt mentions a "car," ChainArchitect infers a suitable context such as a "road." The background description $P_{b_i}$ is passed to the FLUX model by the Coordinator to generate the background image $I_{bg}$, which serves as a spatial reference for placing foreground objects.

To ensure the generated layout follows a consistent, interpretable format (e.g., JSON), ChainArchitect uses in-context exemplars 12 that define object classes, spatial positions, and scene style, thereby aligning the output with the user's intent. Additionally, ChainArchitect leverages GPT-4o's vision capabilities to analyze the background image viewpoint, improving object placement accuracy.

For foreground objects, ChainArchitect performs explicit spatial reasoning: it determines an optimal generation order (placing distant objects before closer ones to manage occlusion) and models inter-object relationships, such as relative positioning ("A is on top of B") and orientation ("Person A is facing left"). This structured reasoning enables coherent and realistic multi-object layouts even in complex scenes.

## 3.3 Object-Integration Network (OIN)

The Object Integration Network (OIN) facilitates the seamless incorporation of objects into pre-existing backgrounds, as illustrated in Figure 3. OIN processes a masked background (delineated by a bounding box), a reference object image, and a text prompt to synthesize a contextually coherent and visually consistent integration of the specified object into the background environment.

**A Parameter Reuse Method for Multiple Conditional Generation via Dual LoRA** Leveraging the robust pretrained capabilities of the FLUX text-image model on text-to-image task, we implement a parameter-efficient adaptation methodology for conditional generation. This approach enables the framework to process masked backgrounds and reference object images for highly precise subject-driven inpainting. Our implementation follows a two-phase training protocol:

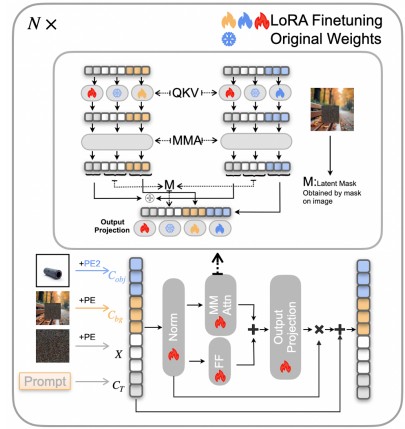

Figure 3: Architecture of the Object Integration Network (OIN). The system processes a text prompt, a background image with a designated bounding box, and a reference object to produce a seamlessly integrated result. Red, yellow, and blue indicators represent the utilization of combined LoRA weights, background inpainting weights, and subject-driven generation weights respectively. "FF" and "MM Attn" denote feedforward layers and multi-modal attention layer in the FLUX model.

In the initial phase, we develop two independent LoRA adaptors—$W_{bg}$ and $W_{obj}$—that enhance the model's capacity to interpret conditional images for inpainting and subject-driven generation tasks. Following the technique proposed in OminiControl [40], we incorporate positional embeddings for background image tokens using encodings identical to the initial noise, while reference image tokens utilize biased embeddings to accommodate spatially aligned and unaligned processing requirements.

The second phase initializes the model with the trained LoRA modules, enabling comprehensive understanding of both background and object conditions ($C_{bg}$ and $C_{obj}$). To circumvent the quadratic memory complexity associated with processing extensive token sequences and to maintain clarity in condition relationships, we bifurcate the latent sequence into two components: $[C_T, X, C_{bg}]$ for background processing and $[C_T, X, C_{obj}]$ for object integration. These components undergo parallel processing with query, key, and value projections utilizing distinct weight sets: $M_{qkv}^{W_{both}}$ for joint conditions, $M_{qkv}$ for FLUX's foundational weights, $M_{qkv}^{W_{inp}}$ for inpainting-specialized LoRA weights,

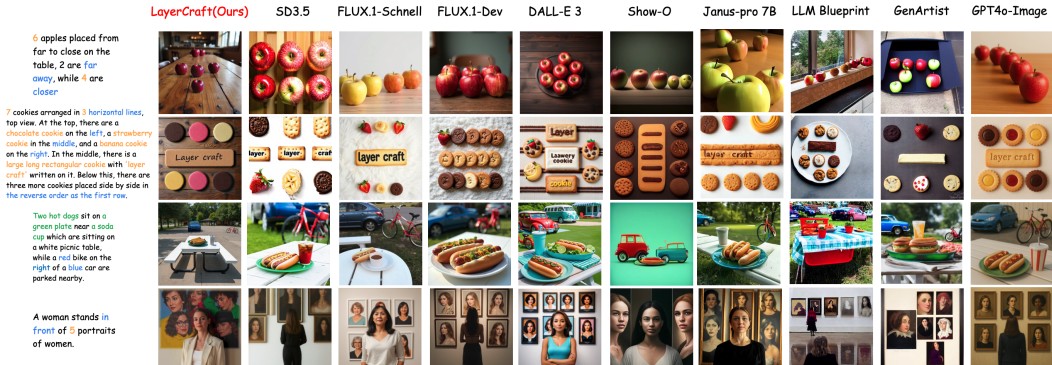

Figure 4: Visual comparisons with state-of-the-art generic text-to-image generation models are presented. On the left, the prompts are annotated with distinct colors to highlight critical attributes and relationships.

and $M_{qkv}^{W_{obj}}$ for object-specific LoRA weights. This architecture generates dual query, key, and value outputs: $[Q_1, K_1, V_1]$ for background elements and $[Q_2, K_2, V_2]$ for object features.

The attention mechanism computes outputs through the following formulations:

$$[C_T^1, X^1, C_{bg}] = \text{Softmax}\left(\frac{Q_1 K_1^T}{\sqrt{d}}\right) V_1,$$

$$[C_T^2, X^2, C_{obj}] = \text{Softmax}\left(\frac{Q_2 K_2^T}{\sqrt{d}}\right) V_2,$$

$$\text{Output} = \left[\frac{C_T^1 + C_T^2}{2}, M(X^1, X^2), C_{bg}, C_{obj}\right],$$

where $M(X^1, X^2)$ denotes the replacement of the masked region's latent sequence $X^1$ with $X^2$ according to the bounding box mask. This methodology preserves both the generative capabilities of the model and its interpretation of the respective conditions, as the resultant image is generated without LoRA layer activation while conditions are processed using their corresponding LoRA weights. Consequently, the training objective focuses on establishing the relationship between textual input and the conditional elements.

**Discussion:** OIN supports LayerCraft's design of the usage of intermediate reference images. It also enables LayerCraft to be an extremely efficient pipeline because during reference image creation, we employ Flux as the primary generator; when reference images are unnecessary, the LayerCraft coordinator can selectively load inpainting LoRA weights. This approach stands in contrast to frameworks such as GenArtist, which necessitates more than ten external models, resulting in computational inefficiency due to model loading/unloading cycles and introducing stylistic inconsistencies stemming from distributional variances across different models. *We will provide more details and analysis related to OIN in the supplementary materials.*

## 4   Experiments

**Implementation Details** We use OpenAI's GPT-4o [1] as the base LLM for both the LayerCraft Coordinator and ChainArchitect agent, with the temperature set to 0.1 to balance control and creativity. Our text-to-image backbone is FLUX.1-dev [18], implemented via the Hugging Face Diffusers library [43].

The Object Integration Network (OIN) is built using Diffusers and PEFT, and trained with a batch size of 1 and gradient accumulation over 4 steps on 4 NVIDIA A6000 Ada GPUs (48GB each). We use a LoRA rank of 4 and enable gradient checkpointing for memory efficiency. OIN is trained for 20,000 iterations on a 50K subset of IPA300K, while OminiControl is fine-tuned for 50,000 iterations. Additional samples are drawn from the remaining dataset for qualitative evaluation.

**Dataset Preparation (IPA300K)** To ensure diversity, we use ChatGPT (via O1) to generate a list of 500 unique objects across various categories. For each object, we create 20 descriptive prompts with

Collage Images Input    LayerCraft(Ours)    LayerCraft (w/o OIN)    GPT-4o    GenArtist (Hacked)

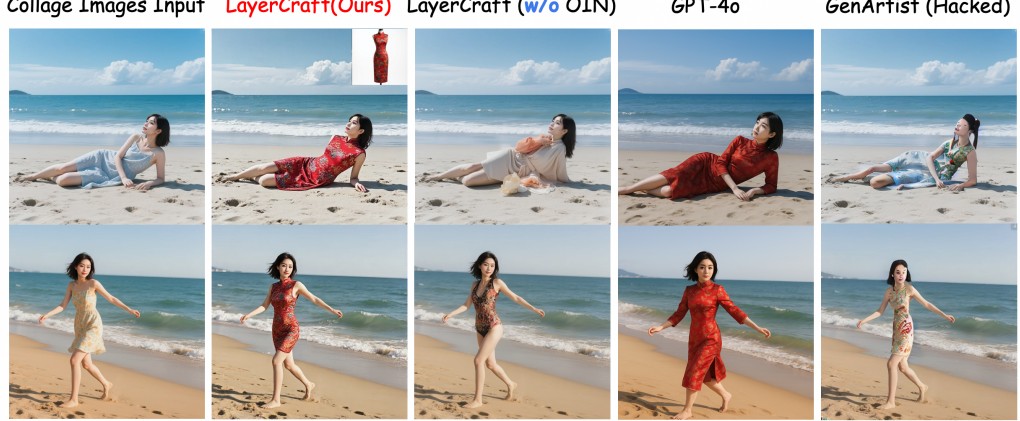

Prompt: Let her wear a Chinese traditional dress.

Figure 5: More example usage of LayerCraft. We can see that our model can generate results with consistent background, and object identity comparing to GPT-4o. It also illustrates the importance of pipeline's design with OIN and intermediate reference images. For GenArtist, even if we provide the grouth truth bounding boxes and extra instructions, they still failed.

varying attributes. Following the procedure in [40], we generate 10 scene-level and 1 studio-level description per prompt to facilitate paired image generation using FLUX.1-dev with 4 random seeds. This results in paired images—one with the object in isolation and one within a complex scene. To obtain accurate object localization, we apply Grounding DINO [24] and SAM 2 [34] to extract bounding boxes from the scene images. Additional image pairs are generated with smaller object sizes to reflect realistic subject-driven inpainting cases in our framework. Bounding boxes are expanded by 15% at the bottom and 10% on each side to reduce the impact of shadows or reflections. After filtering mismatched pairs using LLM-based validation, we obtain a final dataset of 300,000 high-quality pairs, which we name **I**mage-guided in**P**ainting **A**ssets (IPA300K). The dataset will be released on HuggingFace.

Table 1: Comparison with other methods on T2I-Compbench [15]: The ↑ symbol denotes that higher values correspond to better performance. Our LayerCraft system achieves the state-of-the-art performance on the benchmark.

| Method | Attribute Binding | | | Object Relationship | | Numeracy↑ |
|---|---|---|---|---|---|---|
| | Color↑ | Shape↑ | Texture↑ | Spatial↑ | Non-Spatial↑ | |
| LayoutGPT [10] | 0.2921 | 0.3716 | 0.3310 | 0.1153 | 0.2989 | 0.4193 |
| Attn-Exct [5] | 0.6400 | 0.4517 | 0.5963 | 0.1455 | 0.3109 | - |
| GORS [15] | 0.6603 | 0.4785 | 0.6287 | 0.1815 | 0.3193 | - |
| RPG-Diffusion [52] | 0.6024 | 0.4597 | 0.5326 | 0.2115 | 0.3104 | 0.4968 |
| CompAgent [59] | 0.7400 | 0.6305 | 0.7102 | 0.3698 | 0.3104 | - |
| GenArtist [45] | 0.8482 | 0.6948 | 0.7709 | 0.5437 | 0.3346 | - |
| SDXL [32] | 0.6369 | 0.5408 | 0.5637 | 0.2032 | 0.3110 | 0.5145 |
| PixArt-$\alpha$ [7] | 0.6886 | 0.5582 | 0.7044 | 0.2082 | 0.3179 | 0.5001 |
| Playground v2.5 [20] | 0.6381 | 0.4790 | 0.6297 | 0.2062 | 0.3108 | 0.5329 |
| Hunyuan-DiT [22] | 0.6342 | 0.4641 | 0.5328 | 0.2337 | 0.3063 | 0.5153 |
| DALL-E 3 [3] | 0.7785 | 0.6205 | 0.7036 | 0.2865 | 0.3003 | - |
| SD v3 [9] | 0.8085 | 0.5793 | 0.7317 | 0.3144 | 0.3131 | 0.6088 |
| FLUX.1-Dev [19] | 0.7407 | 0.5718 | 0.6922 | 0.2863 | 0.3127 | 0.5872 |
| **LayerCraft (Ours)** | **0.8643** | **0.7046** | **0.8147** | **0.6432** | **0.3508** | **0.6331** |

## 4.1 Visual Comparison with State-of-the-Art Methods

Figure 4 provides a qualitative comparison of our LayerCraft framework against a diverse set of state-of-the-art baselines, including generic diffusion models [19, 3, 9], agent-based approaches [45, 12], autoregressive models [49, 8], and GPT-4o. We crafted prompts that vary in object attributes, quantities, and spatial configurations to rigorously evaluate each method's ability to interpret and realize complex scene compositions.

Our method consistently outperforms competitors by accurately capturing both the object counts and their spatial arrangements. For example, when prompted to generate two apples positioned farther away and four apples closer to the viewpoint, LayerCraft faithfully reproduces the specified quantity and spatial layout. In contrast, while Stable Diffusion 3.5 and FLUX.1-Dev produce the correct number of apples, they fail to preserve the intended spatial relationships. GPT-4o also struggles with correct object counting, and models like PixArt-$\alpha$ and DALL·E 3 frequently generate incorrect object counts. Furthermore, FLUX.1-Schnell and Show-o exhibit notable errors across multiple dimensions, including color, positioning, and object consistency.

Additional visual comparisons are included in the supplementary materials. Figure 5 demonstrates LayerCraft's effectiveness in editing collage photos via a single prompt. Compared to GPT-4o, our framework delivers superior consistency in maintaining coherent backgrounds and faithful human face details, as also illustrated in Figure 6. We further evaluate an ablation without the Object Integration Network (OIN), which forgoes intermediate reference images and results in inconsistent clothing details. Even when using a manually "hacked" version of GenArtist with ground truth bounding boxes and intermediate prompts, the output suffers from blurry faces and inconsistent attire, highlighting the critical role of our intermediate representations and integrated refinement process.

Overall, these results highlight LayerCraft's strengths in robust multi-object control, spatial coherence, and consistent detail preservation, which collectively set it apart from prior approaches.

## 4.2    Comparision on T2I-Compbench

We evaluate our LayerCraft framework against two categories of state-of-the-art approaches: multi-agent systems (upper part) and generic models (lower part) on T2I-Compbench [15] since the GenEval doesn't have the statistics for the agent based models, but we'll include our results in the supplementary materials. As shown in Table 1, LayerCraft excels in all metrics across attribute binding, object relationship, and numeracy, outperforming others due to its instance-level control capabilities.

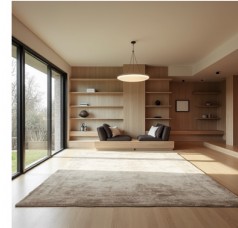

Prompt:
I want to decorate this room.

In contrast to agent-based generation approaches, which typically employ a "generate-then-edit" pipeline, Layer-Craft generates each object sequentially under explicit positional and relational constraints. The generate-then-edit paradigm can propagate early errors into later stages, producing visible artifacts in the final output. Generic diffusion and transformer models fare even worse: lacking the ability to reason over complex textual instructions, they systematically underperform our framework across all evaluated dimensions.

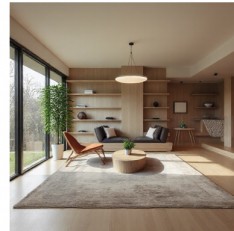

LayerCraft (Ours)

## 4.3    More Quantitative Comparisons

To comprehensively assess the effectiveness of our approach, we conduct extensive quantitative evaluations across multiple benchmarks targeting different aspects of image generation and editing capabilities. First, we evaluate compositional generalization on the GenEval benchmark [13], which systematically tests models' ability to handle complex compositional reasoning through six distinct subcategories (Table 2). LayerCraft achieves an overall score of 0.84, demonstrating strong

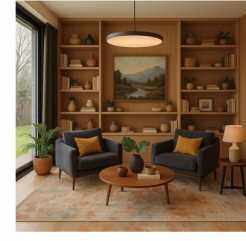

GPT-4o

Figure 6: Another example on indoor decoration, which demonstrates our model's strong consistency.

performance across all evaluated dimensions. Specifically, our model achieves perfect scores (1.0) on Single Object generation, indicating robust object rendering capabilities, and excels in Two Objects composition (0.94), suggesting effective spatial reasoning. The model also shows superior color understanding with a score of 0.89 on the Colors subcategory, while maintaining competitive performance on Counting (0.82) and Position (0.75) tasks. These results notably outperform or match GPT-4o across most subcategories and significantly surpass non-agent baselines including Show-O,

SDXL, DALL-E 3, and SD3.5, highlighting the effectiveness of our layered integration framework for handling complex compositional requirements.

To validate real-world applicability and user preference, we conducted a comprehensive user study with 30 participants evaluating interactive complex prompt generation across diverse scenarios (Table 3). Participants assessed outputs from five state-of-the-art systems—LayerCraft, GPT-4o, LLM-Blueprint, FLUX.1 Dev, and GenArtist—using 15 carefully curated challenging prompts that span various artistic styles, compositional complexities, and semantic requirements. Each generated image was rigorously evaluated on a five-point Likert scale (1 = poor, 5 = excellent) across four critical criteria: (1) prompt consistency, measuring how accurately the generated image reflects all specified elements; (2) naturalness, assessing the realism and coherence of the generated content; (3) visual appeal, evaluating aesthetic qualities including color harmony, compositional balance, and stylistic consistency; and (4) overall quality, providing a holistic assessment. LayerCraft achieved the highest average scores in three out of four criteria, with particularly strong performance in prompt consistency (4.5/5.0) and naturalness (4.5/5.0), surpassing GPT-4o (4.4 and 4.3 respectively). LLM-Blueprint, FLUX.1 Dev, and GenArtist showed consistently lower scores across all categories, with particularly notable deficiencies in prompt consistency and visual appeal, suggesting limitations in their ability to handle complex, multi-faceted generation requirements. Additionally, to demonstrate multi-turn editing capability, we perform multi-round editing tests on the MagicBrush [54] benchmark (Table 4), where our model achieves stronger semantic alignment as reflected by higher CLIP-I/T and DINO scores. These comprehensive evaluations underscore the effectiveness of our structured multi-agent pipeline for complex generation and editing scenarios.

Table 2: Comparison of GenEval results across models. LayerCraft matches or outperforms GPT-4o on most subcategories while significantly surpassing non-agent baselines.

| Model | Overall | Single Obj. | Two Obj. | Counting | Colors | Position | Attr. Bind. |
|---|---|---|---|---|---|---|---|
| LayerCraft | **0.84** | **1.00** | **0.94** | 0.82 | 0.89 | 0.75 | 0.62 |
| GPT-4o | 0.84 | 0.99 | 0.92 | **0.85** | **0.92** | 0.75 | 0.61 |
| Show-O | 0.53 | 0.95 | 0.52 | 0.49 | 0.82 | 0.11 | 0.28 |
| SDXL | 0.55 | 0.98 | 0.74 | 0.39 | 0.85 | 0.15 | 0.23 |
| FLUX.1-Dev | 0.66 | 0.98 | 0.81 | 0.74 | 0.79 | 0.22 | 0.45 |
| DALL-E 3 | 0.67 | 0.96 | 0.87 | 0.47 | 0.83 | 0.43 | 0.45 |
| SD3.5 | 0.71 | 0.98 | 0.89 | 0.73 | 0.83 | 0.34 | 0.47 |
| Janus-pro 7B | 0.80 | 0.99 | 0.89 | 0.59 | 0.90 | **0.79** | **0.66** |

Table 3: Mean ($\mu$) and standard deviation ($\sigma$) of user ratings (1–5). Higher values indicate better performance.

| System | Consistency | Naturalness | Visual Appeal | Overall |
|---|---|---|---|---|
| LayerCraft | **4.5 ± 0.82** | **4.5 ± 0.73** | 4.4 ± 0.78 | **4.2 ± 0.75** |
| GPT-4o | 4.4 ± 0.79 | 4.3 ± 0.86 | **4.5 ± 0.67** | 4.2 ± 0.84 |
| LLM-Blueprint | 3.0 ± 1.28 | 2.9 ± 1.09 | 2.6 ± 1.15 | 2.9 ± 0.94 |
| FLUX.1 Dev | 3.2 ± 1.38 | 3.6 ± 0.97 | 3.5 ± 1.12 | 3.3 ± 1.07 |
| GenArtist | 3.1 ± 1.08 | 3.5 ± 1.18 | 3.6 ± 1.26 | 3.5 ± 1.02 |

Table 4: Multi-turn editing results on MagicBrush. Lower is better for L1/L2; higher is better for CLIP-I, DINO, and CLIP-T.

| Method | L1 ↓ | L2 ↓ | CLIP-I ↑ | DINO ↑ | CLIP-T ↑ |
|---|---|---|---|---|---|
| HIVE | 0.1521 | 0.0557 | 0.8004 | 0.6463 | 0.2673 |
| InstructPix2Pix | 0.1584 | 0.0598 | 0.7924 | 0.6177 | 0.2726 |
| MagicBrush | 0.0964 | 0.0353 | 0.8924 | 0.8273 | 0.2754 |
| GenArtist | **0.0858** | **0.0298** | 0.9071 | 0.8492 | 0.3067 |
| LayerCraft (ours) | 0.0863 | 0.0299 | **0.9121** | **0.8541** | **0.3157** |

## 5 Ablation Study

**Ablation on CoT Variants** To rigorously assess the contribution of Chain-of-Thought (CoT) reasoning in our layout generation process, we conducted a comprehensive ablation study by comparing

Table 5: Ablation Study for CoT on T2I-Compbench.

| Method | Attribute Binding | | | Object Relationship | | Numeracy↑ |
|---|---|---|---|---|---|---|
| | Color↑ | Shape↑ | Texture↑ | Spatial↑ | Non-Spatial↑ | |
| LayerCraft | **0.8643** | **0.7046** | **0.8147** | **0.6432** | **0.3508** | **0.6331** |
| w/o generation order | 0.8524 | 0.6792 | 0.7853 | 0.4210 | 0.3147 | 0.6305 |
| w/o object relationship | 0.8512 | 0.6867 | 0.7842 | 0.4062 | 0.2854 | 0.6301 |
| w/o order & relationship | 0.8413 | 0.6463 | 0.7531 | 0.3847 | 0.2752 | 0.6023 |
| w/o CoT for Layout Generation | 0.6394 | 0.5639 | 0.7216 | 0.2831 | 0.3013 | 0.5663 |

the full LayerCraft pipeline with systematically simplified variants. The purpose is to quantify the individual and collective impact of key CoT components on generation quality and spatial coherence.

Specifically, we evaluated the following variants:

- **Without Generation Order**: Removes the CoT-driven ordering mechanism used to determine the sequence of object placement.
- **Without Object Relationship**: Omits relational reasoning such as spatial prepositions or inter-object dependencies.
- **Without Both Order and Relationship**: Disables both sequential placement and object relationship modeling.
- **Without All CoT for Layout Generation**: Fully removes CoT reasoning from the ChainArchitect, falling back to a single-pass layout prediction without relationships with background.

Due to computational constraints, we employed a stratified sampling strategy and evaluated the models on 20% of the test data, ensuring balanced representation across object types and scene configurations.

As shown in Table 5, the complete LayerCraft pipeline consistently outperforms all ablated versions across key metrics, including object count accuracy, spatial arrangement fidelity, and overall realism. Notably, the absence of generation order and relationship reasoning leads to degraded spatial coherence and increased placement conflicts. The full removal of CoT results in the most significant performance drop, underscoring the critical role of iterative reasoning in managing compositional complexity.

**Limitations** While LayerCraft delivers strong spatial control and compositional accuracy, its use of Chain-of-Thought reasoning and multi-agent coordination introduces additional computational overhead. This can impact efficiency, particularly for complex scenes with many interacting objects. Although spatial accuracy is enhanced through background-guided bounding boxes, the primary cost lies in maintaining agent interactions. Future work will focus on streamlining these processes to improve runtime performance while preserving generation quality.

## 6 Conclusion

We have presented LayerCraft, a novel agent-based framework for text-to-image (T2I) generation that addresses key challenges in compositional control, spatial reasoning, and multi-object fidelity. By integrating three specialized agents, *LayerCraft Coordinator*, *ChainArchitect*, and *Object Integration Network (OIN)*, our system supports structured planning, iterative reasoning, and object-aware image refinement in a fully automated pipeline.

LayerCraft excels in generating complex scenes with accurate spatial layouts and consistent object attributes, all without requiring model finetuning. It also enables consistent multi-image editing, making it particularly effective for tasks such as photo collage editing from a single prompt. Extensive experiments demonstrate superior performance over existing methods in both accuracy and visual coherence.

With instance-level control, real-time interactivity, and a modular design, LayerCraft offers a scalable and user-friendly solution for high-quality image synthesis across a wide range of creative and practical applications.

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

# Appendix

# 7   Broader Impact Statement

LayerCraft significantly advances text-to-image (T2I) generation by providing precise control over composition and object integration, driven by Chain-of-Thought (CoT) reasoning. This research promises substantial positive societal impact, primarily by **democratizing visual content creation** for non-experts, making sophisticated design tools accessible to a broader audience. It will **revolutionize creative and professional workflows** in industries like advertising and gaming, drastically accelerating content creation and fostering innovation. This capability also **catalyzes new forms of digital storytelling and education**, enabling richer visual narratives. While acknowledging risks like misuse for misinformation or bias propagation, which we condemn and will address through ethical guidelines and further research, our core focus remains on LayerCraft's transformative power to empower human creativity and broadly benefit society.

# 8   More Examples on Batch Collage Editing

In this section, we present additional examples of batch collage image editing. Figure 7 illustrates LayerCraft's ability to seamlessly integrate a Van Cleef necklace across multiple photos of a girl. Our model first intelligently identifies optimal placement bounding boxes for the necklace, then generates a consistent reference image to ensure uniformity throughout the process before engaging the Object Integration Network (OIN) for the final result. In contrast, GPT-4o struggles with this task, failing

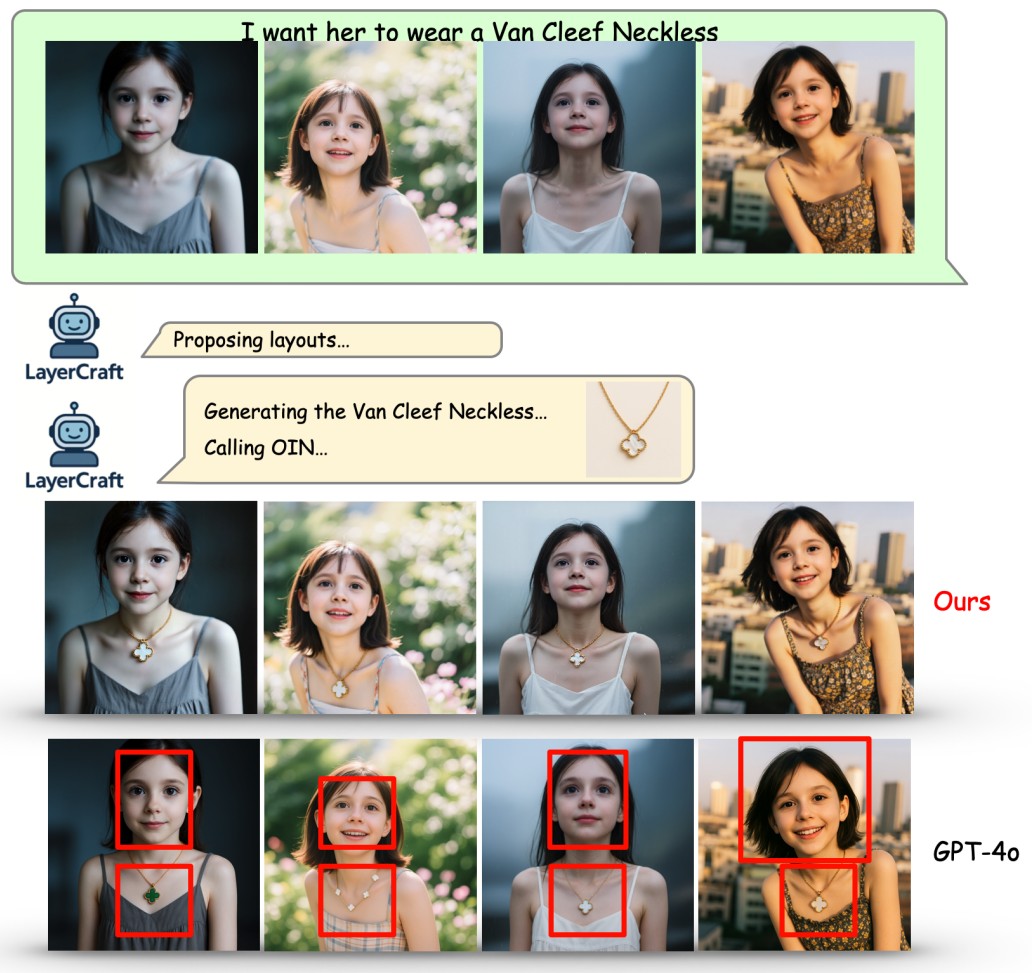

Figure 7: An example of batch collage image editing. LayerCraft effectively proposes bounding boxes for necklace placement and generates a consistent reference image, leading to seamless integration across multiple images with a single prompt. In contrast, GPT-4o fails to preserve facial identity and generates inconsistent necklaces as highlighted by the red boxes.

to preserve facial identity and generating inconsistent necklaces, as highlighted by the red boxes. Figure 8 provides further demonstrations of our model's robust capabilities. The upper panel shows the zoomed-in version of the teaser image. The lower panel showcases a striking outfit modification, seamlessly changing a black man's attire to a white blazer. Figure 9 effectively illustrates the generation of a cohesive Audi advertisement, featuring a single car consistently integrated across five distinct scenes.

# 9 Additional comparisons on T2I generation with other SOTA methods

In this section, Figure 15 presents a detailed qualitative comparison with state-of-the-art methods, including expanded versions of examples from the main paper for clearer visualization. As demonstrated, our model consistently yields better results, particularly in terms of object numeracy and accurate spatial relationships. Furthermore, LayerCraft exhibits significantly fewer artifacts compared to other agent-based methods. For instance, LLM Blueprint generates an anomalous red object beneath the table in the hot dog example. GenArtist, even in its teaser image, struggles with perspective accuracy: while the hotdogs are in focus, the distant car and bike remain sharply defined despite the blurry far end of the table, diminishing overall realism. Our method, conversely, avoids such inconsistencies, producing more coherent and realistic compositions.

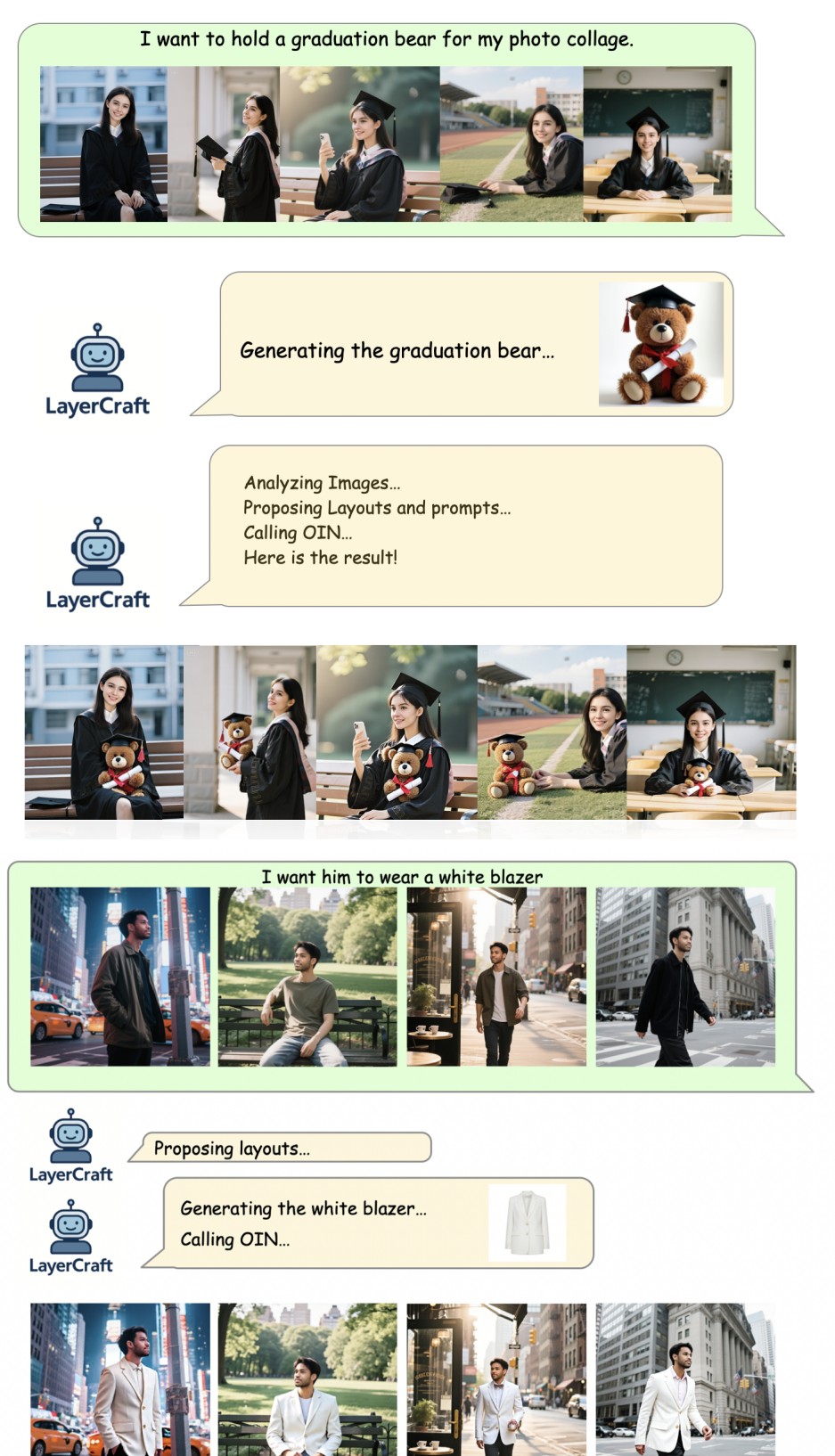

Figure 8: This figure provides more user scenario, the upper part is a larger and clearer demo for the teaser image. The lower panel demonstrates an outfit modification, showing a black man's attire seamlessly changed to a white blazer. These examples highlight the robust capabilities of our model.

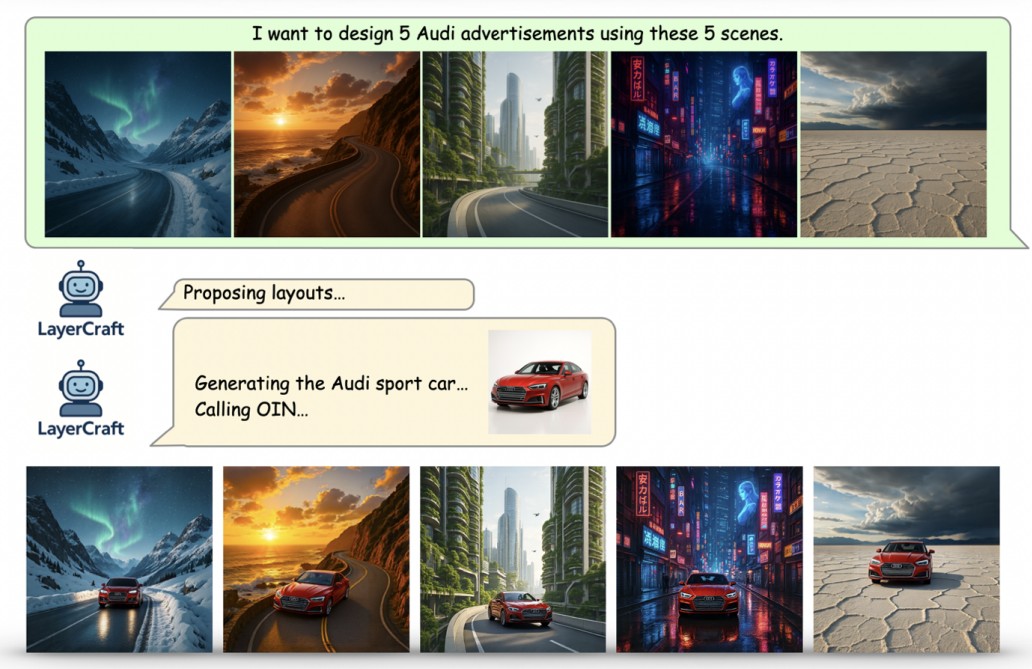

Figure 9: Further examples of batch collage image editing and generation. It illustrates the generation of a consistent Audi advertisement featuring a single car across five distinct scenes.

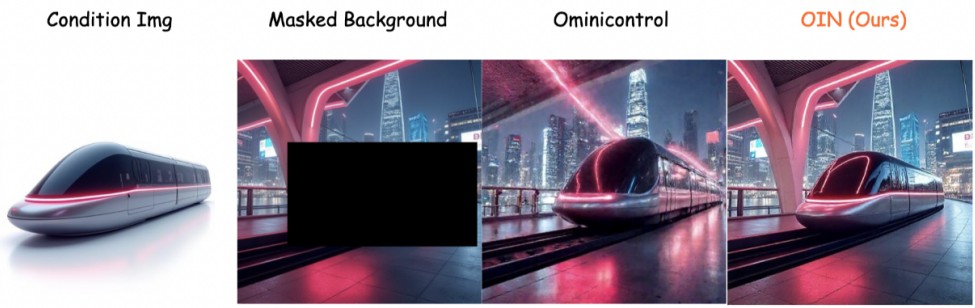

Figure 10: Failure case for direct attention mix in Section 10.1"Analysis of Objection Integration Network". One can see the background is changed and the boundary is easy to see.

## 10 Additional Analysis on Objection Integration Network

### 10.1 Ablation on Attention Mixing

In our work, attention outputs are derived through the strategic blending of dual attention maps utilizing mask indices in the latent space. Specifically, our approach computes attention outputs independently for each branch and subsequently integrates the hidden states according to a latent mask derived from the original masked region. This strategic integration ensures that the model effectively learns optimal object placement within the background while preserving background integrity. Furthermore, since the Multi-Modal attention mechanism within the FLUX architecture processes textual and image tokens concurrently, we address the cross-modal correlations by implementing a weighted average of textual tokens from both branches. This dual-branch integration synergistically enhances the model's comprehension of the conditional inputs. To validate our approach, we conducted comparative analyses against two alternative methodologies. The first alternative, inspired by OmniControl [40], involves extending the input sequence and computing the attention matrix for the entire augmented sequence. However, this methodology encountered significant convergence

challenges during optimization. The second approach implements a weighted summation of attention outputs, which resulted in artifacts characterized by pronounced boundaries in the masked regions and unintended modifications to background elements. Figure 10 presents visual evidence of these failure cases comparing with the successful results obtained using our method. Our proposed methodology demonstrates superior performance compared to these alternatives, as illustrated in more results of OIN in Figures 13 and 14.

## 10.2 Comparisons with Concurrent Subject-driven Inpainting Methods

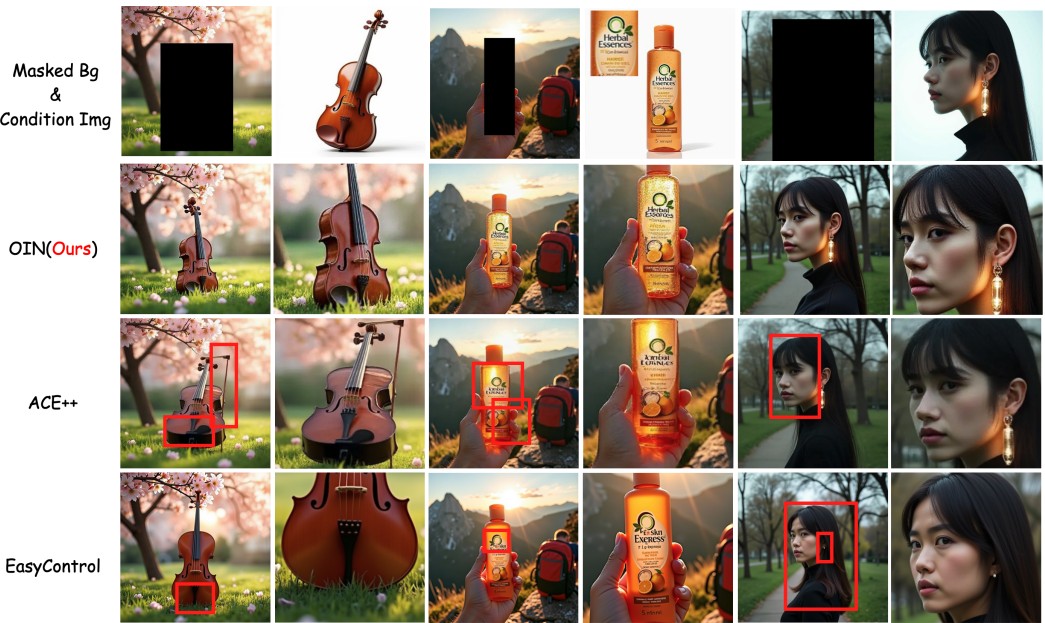

Figure 11: Visual comparisons with concurrent works on subject-driven inpainting task. The first row is our Object-Integration Network, the second row is ACE++ [28], and the last row is EasyControl [56]

In addition, we provide qualitative comparisons with concurrent approaches that support subject-driven inpainting to contextualize our contributions as being on par with, or even surpassing, the current state of the art [28, 56]. Figure 11 displays these results. A detailed examination reveals that our Object Integration Network (OIN) excels in capturing finer details, as exemplified by the orange bottle where text and logos are well-preserved, unlike the blurred text in ACE++'s output or the reverted logo in EasyControl's result. Another instance is the violin, whose features (e.g., the chin rest) are maintained only by OIN, while they disappear in the results from ACE++ and EasyControl. Similarly, the lighted earring on the lady vanishes in EasyControl's output. While these concurrent methods could theoretically replace OIN within our pipeline, OIN offers superior efficiency by requiring only a single loading and unloading of models, as it only necessitates loading the correct LoRA weights. In terms of speed, OIN and ACE++ generate results within one minute, depending on GPU type, whereas EasyControl is two to three times slower, further underscoring OIN's significant contribution to the overall pipeline.

## 11 Handling Difficult Bounding Box Proposal.

To propose a correct bounding box using LLM only will be difficult sometimes, so we adopt a iterative refinement process to solve this problem. Figure 12 demonstrates the process. Initially, the ChainArchitect proposes an initial bounding box. This proposal is then visualized directly on the image, allowing the LLM to iteratively refine its size and position. This refinement process proves highly effective in achieving appropriate bounding box localization.

Bounding Box Refinement...

Bounding Box for Necklace Placement

Further Enlarged Bounding Box for Necklace Placement

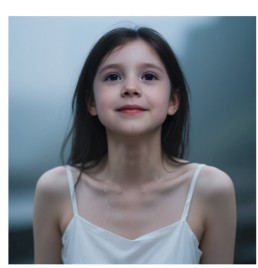 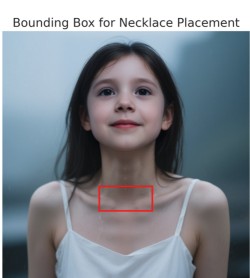 ... 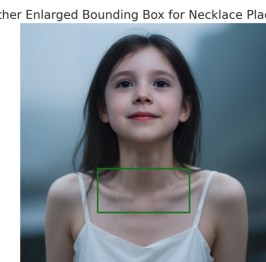

Figure 12: Our approach to handling bounding box proposals in editing tasks demonstrates remarkable effectiveness. Initially, the Large Language Model (ChainArchitect) proposes an initial bounding box. This proposal is then visualized directly on the image, allowing the LLM to iteratively refine its size and position. This refinement process proves highly effective in achieving appropriate bounding box localization.

## 12 Overall Process Decomposed Demo

In this section we will demonstrate how the process happen within the framework. We use the example of living room decoration to demonstrate the CoT thinking process of the framework, it is demonstrated at the end of the supplementary materials.

Fill the blank area in the given background with A patterned cotton scarf for diverse fashion styles., with description: Draped gracefully over a sunlit park bench, it rustles gently in the cool autum...

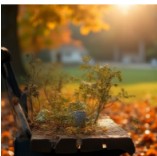 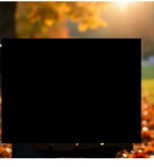 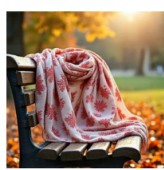

Fill the blank area in the given background with A simple metal measuring scoop., with description: In a brightly lit kitchen, it rests on a marble countertop beside a flour-dusted rolling pin and a m...

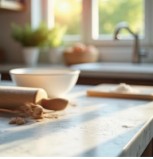 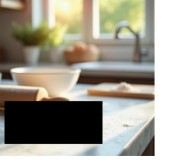 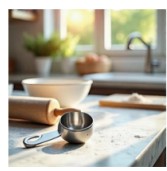

Fill the blank area in the given background with Organizes and safeguards photography gear efficiently., with description: Surrounded by lush green shrubs, it reclines on the vibrant lawn of a country...

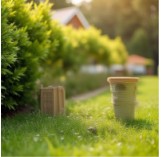 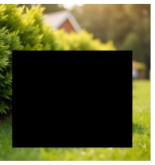 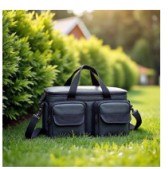

Fill the blank area in the given background with A large, colorful beach shade., with description: Situated on a crowded boardwalk, it provides shade to street performers and curious onlookers, reflec...

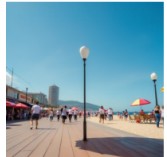 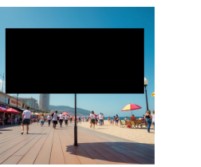 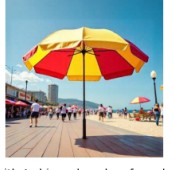 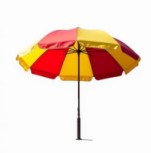

Fill the blank area in the given background with A chic and modern formal outfit., with description: Across a rooftop terrace, it is portrayed in the golden hour light, slightly silhouetted against th...

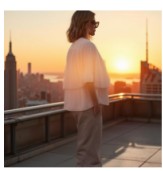 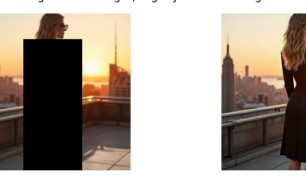 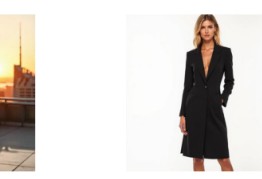

Fill the blank area in the given background with Versatile kitchen appliance with multiple cooking functions., with description: In a modern minimalist kitchen, positioned symmetrically on the island ...

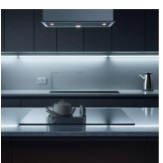 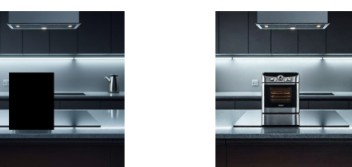 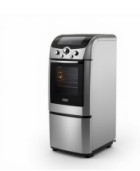

Figure 13: More resluts of OIN

Fill the blank area in the given background with Aircraft engine for Airbus A320 family., with description: In a mechanic's workshop with industrial lighting, it rests on a metal cradle surrounded by ...

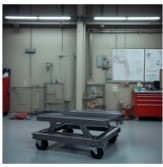 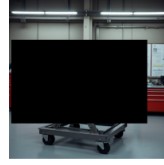 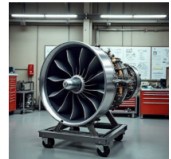 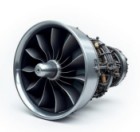

Fill the blank area in the given background with A transparent cup for ceremonial use., with description: Perched on a rustic wooden shelf, it is nestled among weathered books and antique brass items,...

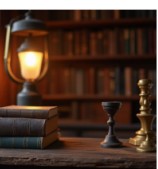 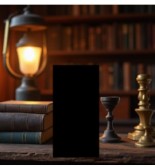 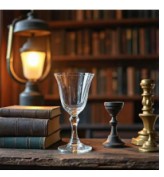 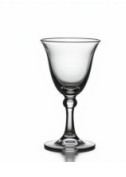

Fill the blank area in the given background with A handcrafted wooden miniature aircraft toy., with description: Perched precariously on a rocky outcrop at golden hour, it catches the last rays of sun...

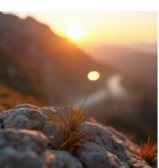 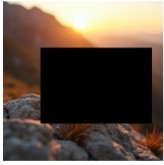 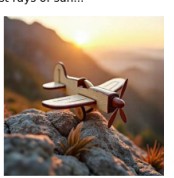 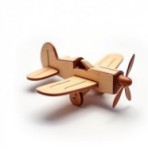

Fill the blank area in the given background with A durable, adventurer-oriented timepiece., with description: Perched on the edge of a rocky shoreline, it rests close to the camera capturing a close-u...

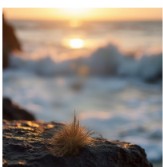 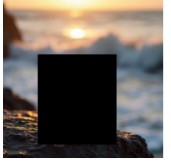 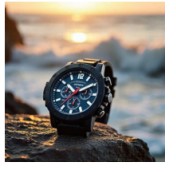 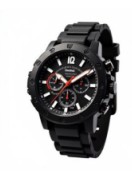

Fill the blank area in the given background with A sleek and elegant pencil skirt., with description: In a quiet library, a side view shows off its refined structure under gentle yellow lamps, surroun...

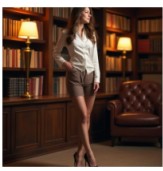 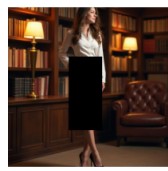 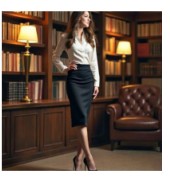 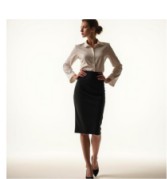

Fill the blank area in the given background with A sturdy wooden bunk bed for youth., with description: Nestled in a cozy cabin bedroom, this item is captured from a side angle with the warm glow of a...

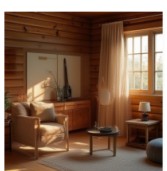 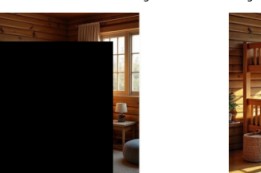 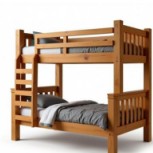

Figure 14: More results of OIN

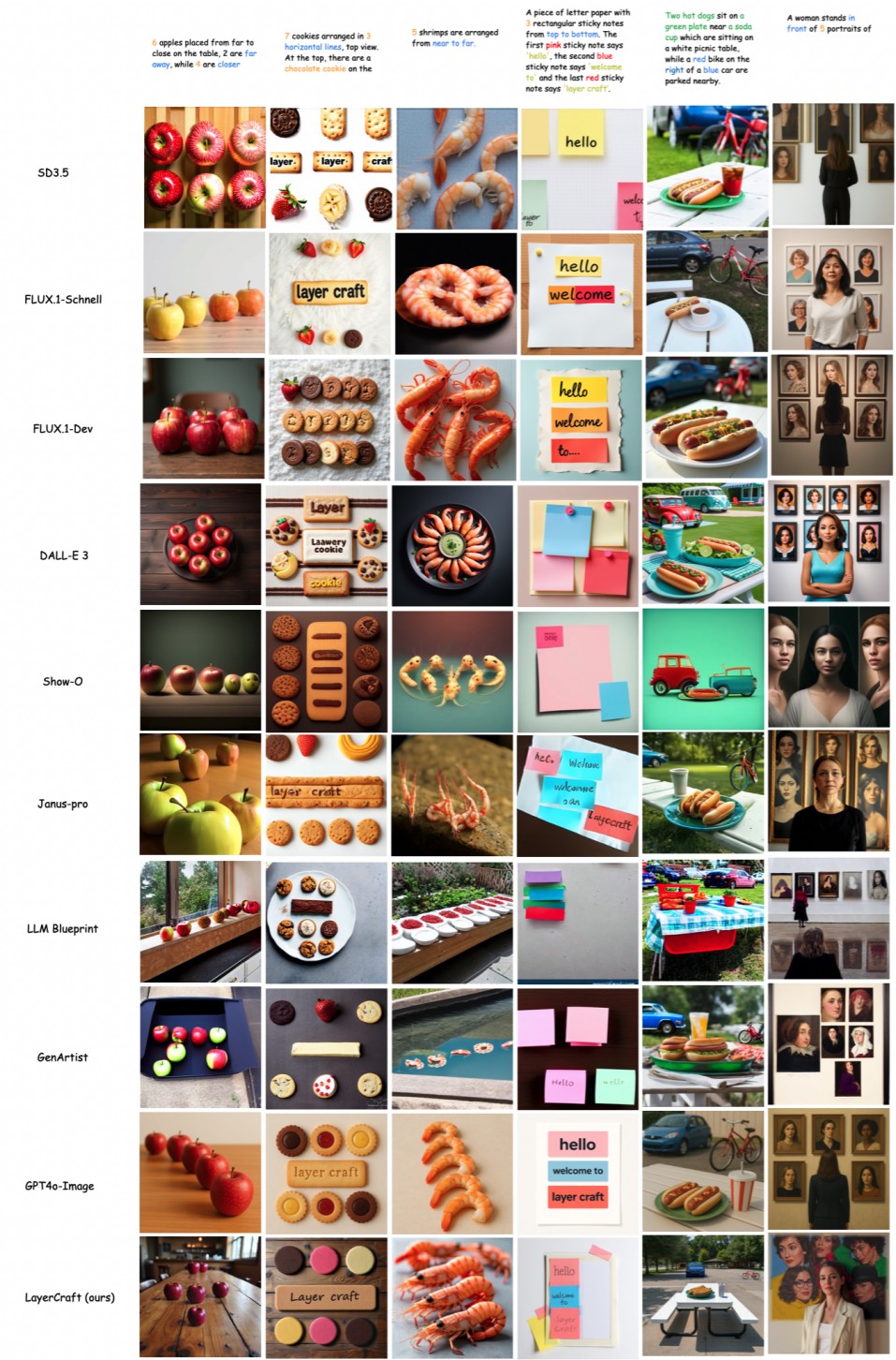

Figure 15: More comparisons with state-of-the-art models. We also include the examples used in the main paper and make them larger for details.

USER: I want to decorate this room.

LAYERCRAFT COT THINKING PROCESS: This image shows a minimalist, modern living
room with a clean, warm aesthetic. Let me identify the objects and describe the
background in JSON format.

```json
{
  "objects": [
    {
      "name": "reading nook",
      "description": "A built-in wooden seating area with
          dark cushions and white pillows nestled within
          the wooden shelving unit"
    },
    {
      "name": "pendant light",
      "description": "A circular, disc-shaped hanging
          light fixture with a diffuse glow suspended
          from the ceiling"
    },
    {
      "name": "area rug",
      "description": "A large, neutral-toned rectangular
          rug covering part of the wooden floor"
    },
    {
      "name": "shelving unit",
      "description": "Built-in light wood shelving
          spanning the back wall with various decorative
          items displayed"
    },
    {
      "name": "decorative objects",
      "description": "Small sculptures, books, and framed
          artwork arranged sparsely on the shelves"
    },
    {
      "name": "sliding glass doors",
      "description": "Large floor-to-ceiling glass doors/
          windows on the left side offering views to the
          outdoors"
    }
  ],
  "background": {
    "description": "A spacious, minimalist room with
        light wooden flooring, white ceiling, and light
        wood wall paneling. The room features built-in
        shelving along the back wall and large sliding
        glass doors that allow natural light to flood the
         space. The viewpoint is from the center of the
        room looking toward the back wall with the built-
        in shelving and seating area, with the glass
        doors visible on the left side.",
    "included_elements": ["wooden flooring", "white
        ceiling", "light wood wall paneling", "natural
        lighting"]
  }
}
```

The shelving unit and built-in seating are considered separate objects but could be
viewed as part of the architectural design. I've included the structural elements like
flooring, ceiling, and wall paneling in the background description since they define
the space rather than being foreground objects.

Looking at the image, I can suggest several items that could be added to enhance the minimalist space while maintaining its clean aesthetic. Let me think about what would complement this room:

```
{
  "suggested_additions": [
    {
      "name": "coffee table",
      "description": "A low, rectangular wooden coffee
          table with clean lines, placed in front of the
          reading nook on the area rug"
    },
    {
      "name": "side table",
      "description": "A small, round side table in light
          wood or brass next to the reading nook"
    },
    {
      "name": "floor lamp",
      "description": "A slim, architectural floor lamp
          with warm lighting placed in the corner
          opposite the glass doors"
    },
    {
      "name": "accent chair",
      "description": "A minimalist armchair in a
          complementary neutral tone (tan leather or
          light gray fabric) positioned at an angle to
          the reading nook"
    },
    {
      "name": "throw blanket",
      "description": "A textured throw blanket in a
          subtle pattern or muted color draped over one
          corner of the reading nook"
    },
    {
      "name": "indoor plant",
      "description": "A tall potted plant like a fiddle
          leaf fig or snake plant placed near the glass
          doors to bring in natural elements"
    },
    {
      "name": "decorative bowl",
      "description": "A sculptural bowl or vessel on one
          of the open shelves to add visual interest"
    }
  ]
}
```

These additions would enhance the functionality and warmth of the space while respecting its minimalist design language and natural materials.

Analyzing the image based on the provided grid regions, I'll determine suitable placement areas for the suggested objects while considering perspective and available space:

```
{
  "object_placements": [
    {
      "object": "coffee table",
      "suitable_regions": ["center", "between center and
          center-right", "between lower center and center
          ", "between lower center-right and center"],
```

```json
      "reasoning": "The central floor area on the rug
          provides ample space for a coffee table,
          positioned in front of the reading nook"
    },
    {
      "object": "side table",
      "suitable_regions": ["center-right", "between
          center and center-right"],
      "reasoning": "A small side table would fit well
          beside the reading nook on the right side"
    },
    {
      "object": "floor lamp",
      "suitable_regions": ["lower left", "between lower
          left and center-left"],
      "reasoning": "The corner area opposite the glass
          doors provides good placement for a tall floor
          lamp without blocking views or pathways"
    },
    {
      "object": "accent chair",
      "suitable_regions": ["lower center-left", "between
          lower center-left and center", "center-left", "
          between center-left and center"],
      "reasoning": "An accent chair would fit well in
          this area, creating a conversation zone with
          the reading nook while maintaining open space"
    },
    {
      "object": "throw blanket",
      "suitable_regions": ["center-right"],
      "reasoning": "The throw blanket would be placed on
          the reading nook which is already in this
          region"
    },
    {
      "object": "indoor plant",
      "suitable_regions": ["upper left", "between upper
          left and center-left"],
      "reasoning": "Near the glass doors to receive
          natural light while adding greenery to that
          corner of the room"
    },
    {
      "object": "wall art",
      "suitable_regions": ["upper center-left", "upper
          center", "between upper center-left and center"
          , "between upper center and center"],
      "reasoning": "The visible wall space opposite the
          shelving unit would accommodate wall art while
          maintaining visual balance"
    },
    {
      "object": "decorative bowl",
      "suitable_regions": ["center-right", "between
          center and center-right"],
      "reasoning": "Could be placed on one of the
          existing shelves in the built-in unit"
    }
  ],
  "spatial_considerations": {
    "viewing_perspective": "The image is taken from a
        position looking toward the back wall with the
        built-in shelving, with the glass doors on the
        left side",
```

```
        "floor_space": "The central and lower areas have the
            most available floor space for furniture
            placement",
        "traffic_flow": "Pathways should be maintained
            between the entrance (presumed to be behind the
            viewing position) and the glass doors",
        "focal_points": "The reading nook and shelving unit
            already serve as focal points, so additional
            elements should complement rather than compete
            with them"
    }
}
```

This analysis considers both the physical space available in each region and the design principles of balance, flow, and functionality while maintaining the minimalist aesthetic of the room.

Based on our analysis, here are the bounding boxes and prompts for each suggested object, ordered from far to close in terms of generation order:

```
{
  "objects": [
    {
      "type": "indoor plant",
      "position": "upper left",
      "generation_order": 1,
      "prompt": "A tall, elegant fiddle leaf fig plant
          with large, glossy green leaves in a minimalist
           ceramic pot, placed near large windows to
          receive natural light, rendered in a clean,
          contemporary style that complements modern
          interior design.",
      "bounding_box": [50, 150, 150, 350]
    },
    {
      "type": "decorative bowl",
      "position": "center-right",
      "generation_order": 2,
      "prompt": "A sculptural, handcrafted ceramic bowl
          in a soft matte finish with subtle organic
          texture and asymmetrical form, placed on an
          open shelf, rendered in a minimalist
          Scandinavian style.",
      "bounding_box": [420, 220, 470, 260]
    },
    {
      "type": "throw blanket",
      "position": "center-right",
      "generation_order": 3,
      "prompt": "A soft, textured throw blanket in muted
          beige with subtle geometric pattern, casually
          draped over the corner of the built-in seating
          area, rendered in a warm, inviting style that
          adds comfort to the minimalist space.",
      "bounding_box": [420, 250, 500, 300]
    },
    {
      "type": "side table",
      "position": "between center and center-right",
      "generation_order": 4,
      "prompt": "A small, round side table with slender
          brass legs and a light oak top, positioned
          beside the reading nook at perfect arm's reach
          height, rendered in a clean, architectural
          style.",
```

```
      "bounding_box": [350, 250, 420, 320]
    },
    {
      "type": "floor lamp",
      "position": "between lower left and center-left",
      "generation_order": 5,
      "prompt": "A tall, architectural floor lamp with a
          slender brushed brass stem and minimal white
          shade casting a warm glow, positioned in the
          corner of the room, rendered in a contemporary
          style that emphasizes clean lines.",
      "bounding_box": [70, 320, 150, 450]
    },
    {
      "type": "accent chair",
      "position": "center-left to between center-left and
          center",
      "generation_order": 6,
      "prompt": "A minimalist lounge chair with gentle
          curves, upholstered in natural tan leather with
           a light wooden frame, positioned at an
          inviting angle to create conversation space,
          rendered in a Scandinavian modern style.",
      "bounding_box": [100, 250, 250, 380]
    },
    {
      "type": "coffee table",
      "position": "center to between lower center and
          center",
      "generation_order": 7,
      "prompt": "A low, rectangular coffee table with
          clean lines and rounded corners, crafted from
          light oak with a subtle grain pattern and
          minimalist design, positioned centrally on the
          area rug, rendered in a contemporary style that
           balances form and function.",
      "bounding_box": [200, 300, 350, 400]
    }
  ]
}
```

This arrangement considers depth perception (generating background objects first),
spatial relationships, and the existing elements in the room. The bounding boxes
are designed to place objects in appropriate locations while maintaining a balanced,
functional layout that enhances the minimalist aesthetic of the space.

USER: I've done the previous decorating and want to put this cute teddy bear in.

LAYERCRAFT: Looking at the new item you want to add to the room, I can help place
the teddy bear in the previously discussed space.

```
{
  "additional_object": [
    {
      "type": "teddy bear",
      "position": "lower center-right",
      "generation_order": 8,
      "prompt": "A small, golden-brown teddy bear with a
          smiling face and soft plush texture, sitting
          upright on one of the built-in shelves, adding
          a touch of warmth and playfulness to the
          minimalist space.",
      "bounding_box": [290, 300, 480, 490]
    }
  ]
```

```
}
```

These placements would work well with your previous decorating plans.

