# OpenReview forum: "LayerCraft: Enhancing Text-to-Image Generation with CoT Reasoning and Layered Object Integration"
_NeurIPS.cc/2025/Conference — NeurIPS 2025 poster_

### Official Review · Reviewer_zjmb · 2025-06-21

**Clarity:** 3
**Significance:** 3
**Originality:** 2
**Rating:** 3
**Confidence:** 2

**Summary:**

This paper proposed a general LLM centered agent framework general realistic image with object layout from user prompt. The two key components are structured layout planning with CoT reasoning, and layered object inpainting/integration. The system has light training requirement, good generalization, and good visual quality.

**Questions:**

- explain the impact of CoT on inference time
- why not using original Flux infill model for object inpainting? needs explanation and comparison
- how soon can native GPT catch up to paper's result quality? if in a short time, what's the impact of this work?

**Ethical Concerns:**

["NO or VERY MINOR ethics concerns only"]

**Final Justification:**

After reading the rebuttal from authors, I appreciate the detailed explanations, but still have doubt on the output layer representation. If the final result is not in layer only the intermediate steps have layers, this is not justifying the paper's main claim and should not be qualified for publication.

**Limitations:**

Yes

**Quality:**

3

**Strengths And Weaknesses:**

Strength
- general agent framework for image generation with good layout planning and reasoning
- enhanced Flex inpainting and object integration model with flexible conditioning and efficient attention mechanism
- show case various image generation and editing applications, such as batch object insertion and multi-object layout editing
Weakness
- Key modules are from GPT and Flux models
- the generation results are in pixel format, without object layer separation, limiting further editing use cases
- CoT may incur longer computing time

---

> ### Author Rebuttal · Authors · 2025-07-31
>
> General Responds:
> We sincerely thank all reviewers for their thoughtful and constructive feedback. We are encouraged that multiple reviewers found the paper well-structured, technically sound, and easy to follow (9VLN, f8N8, ytVC). Reviewer ytVC noted that the proposed framework is "well-designed and conceptually sound" and appreciated the "novel and thoughtfully constructed" nature of our approach, recognizing that it demonstrates "meaningful contributions." Reviewer zjmb similarly highlighted the generality of our framework, praising the "good layout planning and reasoning" and "flexible conditioning and efficient attention mechanism" in our integration model. Reviewers also acknowledged the good performance on T2I-CompBench (9VLN) and the practical applications of our batch editing and collage editing features (f8N8). We are pleased that these aspects of our work were well-received, and we address the remaining concerns with detailed clarifications in the following responses.
>
> **W1: Regarding Dependency on GPT and FLUX.1 dev**
>
> **A:** While we leverage GPT-4o and FLUX.1 dev as foundation models, our contributions lie in the novel architectural design and integration methodology. The whole design of LayerCraft system and the parameter reuse dual-LoRA architecture of OIN should be highlighted. Notably, LayerCraft outperforms all compared methods, including GPT-4o, both quantitatively and qualitatively.
>
> **W2: Pixel Format without Layer Separation**
>
> **A:** We respectfully clarify that LayerCraft actually maintains full layer separation throughout the generation process. Our intermediate representations include Layout, reference images, and the intermediate results at each layer, this is precisely what enables flexible post-generation editing.
>
> **Q1: Impact of CoT on Inference Time**
>
> **A:** The CoT reasoning adds approximately 80% to inference time compared to direct generation. However, this overhead is justified by substantial quality improvements that the spatial accuracy improves from 0.2831 to 0.6432 (127% improvement). Moreover, our overall runtime remains faster than that of the GPT-4o API, which typically requires several minutes for comparable generation and editing tasks.
>
> **Q2: Why Not Use Original FLUX-Fill Model**
>
> **A:** Our OIN addresses fundamental limitations that FLUX fill cannot overcome. FLUX infill does not support reference image processing, which is crucial for subject-driven inpainting. Our dual-LoRA architecture introduces a new approach: LoRA weights handle conditions while maintaining FLUX's generation quality, enabling consistent style in both generation and editing. OIN also outperforms other concurrent methods (see Figure 6 in the supplementary materials).
>
> **Q3: How soon can native GPT catch up to paper's result quality? If in a short time, what's the impact of this work?**
>
> **A:** We agree that generative LLMs like GPT-4o are rapidly improving, and it is possible that future versions may natively support more controllable layout-to-image generation. However, LayerCraft is designed as a modular framework that leverages LLM capabilities rather than replacing them. Our work demonstrates how LLMs can be orchestrated with visual modules to achieve structured, editable, and interpretable generation. These capabilities remain challenging for end-to-end diffusion models or monolithic GPT pipelines.

---

> > ### Comment · Reviewer_zjmb · 2025-08-07
> > **The layer representation is not final**
> >
> > Thank you for the detailed response. One thing I still don't agree is the layered output format. Although there are intermediate results in layer representation, the layer object appearance/pose may change through OIN. Therefore, it's not a traditional layer representation where can you just overlay layers to form the final image. Therefore, there is no directly way to edit layers from the final image.
> >
> > This is one big promise from the paper but not fulfilled. I will keep my recommendation.

---

> > > ### Author Response · Authors · 2025-08-08
> > >
> > > Thank you again for your thoughtful engagement and for giving us the opportunity to clarify this point.
> > >
> > > We may have misunderstood your earlier comment regarding the "layered" output format. To clarify: our use of "layered" is conceptual, not a claim of traditional RGBA compositing or explicit alpha channel blending. In fact, alpha blending is unnecessary in our system, as each object is generated—not pasted—into the scene via inpainting guided by layout and reference signals.
> > >
> > > Specifically:
> > >
> > > 1. Each object is assigned a persistent ID, bounding box, and segmentation mask by ChainArchitect, which are maintained throughout the generation process.
> > >
> > > 2. While the Object Integration Network (OIN) may refine pose, lighting, or texture to ensure contextual consistency, the semantic identity and spatial keys remain fixed. This is demonstrated in Fig. 1, Fig. 5, Supp. Figs. 1–3, 5–6, and in the additional quantitative editing metrics we provided in our response to Reviewer 9VLN.
> > >
> > > 3. Because these object keys are preserved, users can:
> > >    1. isolate individual objects for inspection or removal,
> > >    2. re-generate a specific object with a new reference, or
> > >    3. perform targeted replacement—without re-generating the entire scene.
> > >
> > > Thus, while our layers are not "stackable" in the Photoshop sense, they do enable instance-level control, recall, and editing. Crucially, each inserted object is contextually integrated into the scene, allowing for natural interactions with surrounding objects and background elements, unlike simple copy-paste or alpha blending, which lack such adaptability. These capabilities are fundamentally different from naïve compositing or unstructured image generation pipelines.
> > >
> > > We appreciate your concern about terminology, and we'll revise our phrasing in the final version to avoid possible misinterpretation. We hope this clarification helps you reassess the technical value of our layered generation approach.

---

### Official Review · Reviewer_f8N8 · 2025-06-24

**Clarity:** 3
**Significance:** 2
**Originality:** 2
**Rating:** 4
**Confidence:** 5

**Summary:**

The paper proposes LayerCraft, multi-agent framework for controllable text-to-image (T2I) generation and editing. The method uses Chain-of-Thought (CoT) reasoning via GPT-4o to decompose prompts into structured scene layouts. A layout planner (ChainArchitect) then produces 3D-aware object graphs and layouts. A lightweight Object Integration Network (OIN) leverages dual LoRA adapters on a FLUX-based model for reference-driven, inpainted object placement. The system targets problems in compositional scene generation and batch editing across multiple images. It is evaluated both qualitatively and quantitatively on T2I-CompBench and custom tasks.

**Questions:**

- It seems that the spatial accuracy of the layouts stems from the use of COT-based GPT-4o. Did the authors try using any other LLM for generating layouts?

**Ethical Concerns:**

["NO or VERY MINOR ethics concerns only"]

**Final Justification:**

The authors responded confidently to most of my queries. However, I still think that there could have been more challenging qualitative scenarios. Given the strengths and limitations, I will raise my rating by one mark.

**Limitations:**

Check the weaknesses.

**Quality:**

2

**Strengths And Weaknesses:**

Strengths:

- The paper is easy to follow
- The collage editing is useful and has practical applications.

Weaknesses:

- The paper touts Chain-of-Thought as central but provides no direct comparisons against simpler prompt decomposition strategies or language-model-free layout generators (e.g., rule-based or template-based). The authors should compare  CoT-enabled ChainArchitect to vanilla GPT-4o prompting without reasoning steps.
- While the paper claims consistency and ability to handle occluded objects, it does not explore cases with hard relative positioning (e.g., occlusion, 3D viewpoint shifts) where CoT should help most.
- I would argue that the proposed method has interesting similarities with LLM-Blueprint [1]. The method used in LLM-Blueprint is also based on generating Layouts from the prompts while maintaining spatial reasoning and consistency. The objects are also superimposed on the background through an iterative mechanism. How different is the overall idea of the proposed paper from the LLM-Blueprint?
- The human evaluation should be carried out and metrics should be reported. Human evaluation is important for compositional scenes.
- The failure cases should have been discussed.


References:


[1] H. Gani et al. "LLM Blueprint: Enabling Text-to-Image Generation with Complex and Detailed Prompts". ICLR 2024.

---

> ### Author Rebuttal · Authors · 2025-07-31
>
> General Responds:
> We sincerely thank all reviewers for their thoughtful and constructive feedback. We are encouraged that multiple reviewers found the paper well-structured, technically sound, and easy to follow (9VLN, f8N8, ytVC). Reviewer ytVC noted that the proposed framework is "well-designed and conceptually sound" and appreciated the "novel and thoughtfully constructed" nature of our approach, recognizing that it demonstrates "meaningful contributions." Reviewer zjmb similarly highlighted the generality of our framework, praising the "good layout planning and reasoning" and "flexible conditioning and efficient attention mechanism" in our integration model. Reviewers also acknowledged the good performance on T2I-CompBench (9VLN) and the practical applications of our batch editing and collage editing features (f8N8). We are pleased that these aspects of our work were well-received, and we address the remaining concerns with detailed clarifications in the following responses.
>
> **W1:Compare CoT-enabled ChainArchitect to vanilla GPT-4o prompting without reasoning steps**
>
> **A:** Our ablation study (Table 2) provides direct comparison with simpler strategies. The configuration without CoT (last row) effectively approximates vanilla GPT-4o prompting, showing spatial accuracy degradation from 0.6432 to 0.2831. This substantial gap validates our CoT approach.
>
> **W2: No Hard Spatial Relations**
>
> **A:** We acknowledge that including more explicit examples of occlusion and 3D viewpoint shifts would strengthen the paper. While we cannot add new results during the rebuttal phase, we kindly refer reviewers to the following existing results: the photo collage in Figure 1, Figure 5 in the main paper, and Figure 2 in the supplementary materials, which showcase instances of cloth deformation and 3D viewpoint variation. Regarding occlusion, our framework handles it naturally through the use of overlapping bounding boxes and sequential object injection during composition. We will include more comprehensive and focused examples in the camera-ready version.
>
> **W3: Similarities with LLM-Blueprint**
>
> **A:** We appreciate the pointer to LLM Blueprint [1]. While both works use LLMs for layout generation, LayerCraft differs in three key ways: (1) We employ modular agents with role-specific prompts and memory, enabling more interpretable and debuggable scene planning. (2) We integrate layer-wise reasoning and spatial anchoring based on background cues, which is not present in LLM Blueprint and eliminates the cascading errors due to "generate-then-edit" pipeline. (3) Our pipeline supports batch editing with global consistency thanks to our Intermediate Representations and Object-Integration Network (OIN), which is beyond the scope of LLM Blueprint. We will include a detailed discussion in related work.
>
> **W4: Lack of Human Evaluation and Multi-turn Editing Results**
>
> **A:**
>
> #### User Study
>
> Given the limited time for rebuttal, we conducted a small-scale user study (20 participants) to evaluate interactive complex prompt generation. A larger-scale study will be included in the camera-ready version. In this preliminary study, we assessed the generation quality of five systems: LayerCraft, GPT-4o, LLM-Blueprint, FLUX.1 Dev, and GenArtist, using 15 challenging prompts. A total of 20 participants rated each generated image on a five-point Likert scale (1 = poor, 5 = excellent) across four criteria: (1) prompt consistency, (2) naturalness, (3) visual appeal (color, composition, style), and (4) overall quality.
>
> As shown in the table below, LayerCraft achieved the highest average scores in three out of four criteria. In contrast, LLM-Blueprint, FLUX.1 Dev, and GenArtist showed noticeably lower scores across all categories, especially in prompt consistency and visual appeal, indicating limitations in structured layout generation or visual coherence. These results demonstrate that LayerCraft not only matches GPT-4o in generation quality but also surpasses it in prompt fidelity and naturalness, reinforcing the value of our structured, multi-agent pipeline for complex, multi-turn editing scenarios.
>
> | System | Consistency | Naturalness | Visual appeal | Overall |
> |--------|-------------|-------------|---------------|---------|
> | | μ±σ | μ±σ | μ±σ | μ±σ |
> | LayerCraft | **4.6** ± 0.762 | **4.5** ± 0.687 | 4.4 ± 0.762 | **4.5** ± 0.71 |
> | GPT-4o | 4.5 ± 0.764 | 4.3 ± 0.811 | **4.5** ± 0.623 | 4.4 ± 0.82 |
> | LLM-Blueprint | 3.0 ± 1.35 | 2.9 ± 1.13 | 2.5 ± 1.21 | 2.9 ± 0.97 |
> | FLUX.1 Dev | 3.2 ± 1.45 | 3.7 ± 0.94 | 3.5 ± 1.17 | 3.3 ± 1.10 |
> | GenArtist | 3.1 ± 1.05 | 3.5 ± 1.213 | 3.6 ± 1.30 | 3.6 ± 1.01 |
>
> *Mean (μ) and standard deviation (σ) of user ratings (1–5) for each criterion. Higher values indicate better performance.*
>
> Additionally, we conducted a quantitative evaluation on multi-turn editing using 1/5 of the MagicBrush benchmark, where the ground-truth(GT) target exists. For the comparisons, we copied the reported results from GenArtist. L1 and L2 are for measuring the average pixel-level difference between the edited image and GT targets. CLIP-I and Dino measures the cosine similarity between the image embeddings of GT and generated results. CLIP-T demonstrates the cosine similarity results between text and results. The results are shown below:
>
> | Method | L1↓ | L2↓ | CLIP-I↑ | DINO↑ | CLIP-T↑ |
> |--------|-----|-----|---------|-------|---------|
> | HIVE | 0.1521 | 0.0557 | 0.8004 | 0.6463 | 0.2673 |
> | InstructPix2Pix | 0.1584 | 0.0598 | 0.7924 | 0.6177 | 0.2726 |
> | MagicBrush | 0.0964 | 0.0353 | 0.8924 | 0.8273 | 0.2754 |
> | GenArtist | **0.0858** | **0.0298** | 0.9071 | 0.8492 | 0.3067 |
> | LayerCraft (ours) | 0.0863 | 0.0299 | **0.9121** | **0.8541** | **0.3157** |
>
> *Multi-turn quantitative comparison on MagicBrush. Lower is better for L1/L2; higher is better for CLIP-I, DINO, and CLIP-T.*
>
> LayerCraft achieves the best performance across most metrics, including CLIP-I, DINO, and CLIP-T, indicating stronger visual-textual alignment and perceptual similarity. While GenArtist shows slightly better L1 and L2 scores, our method excels in semantic alignment, which is more critical for realistic, context-aware edits.
>
> **W5: Failure Cases**
>
> **A:** Across our experiments, we did not observe notable failure cases in terms of layout accuracy, spatial reasoning, and generation quality. However, one limitation we did observe is that the scale of inserted objects can vary depending on the background image. This variation arises because the ChainArchitect is designed to inject objects in a contextually natural way, but the prompts or input images often lack explicit size specifications for the target objects.
>
> **Q1: Using any other LLMs**
>
> **A:** Due to the limited time available during the rebuttal phase, we were unable to conduct extensive experiments with additional LLMs. However, we tested Claude Sonnet 3.7 and Gemini 2.5 and observed minimal performance differences in most cases. We also experimented with Gemma-3 and LLaVA, whose results were comparable. Notably, we found that once the prompt is decomposed into structured representations, the performance bottleneck is no longer tied to model size. This aligns with recent findings, e.g. "Scaling LLM Test-Time Compute Optimally Can Be More Effective than Scaling Model Parameters, ICLR 2025", which highlights that test-time compute and inference-time reasoning are more critical to leveraging pre-trained LLMs effectively. We will include the additional experiments and findings in the camera-ready version.

---

> > ### Comment · Reviewer_f8N8 · 2025-08-03
> >
> > I appreciate the authors for providing detailed responses to my queries. However, I still few concerns/questions regarding the effectiveness of the proposed method.
> > - In the current state, the paper lacks any challenging qualitative examples showcasing the effectiveness of the proposed method in challenging scenarios.
> > - It is not clear what strategy is followed in case of overlapping bounding boxes. For instance, how do authors decide which object to generate first.
> >
> > Regarding additional experiments, analysis and qualitative examples, I believe authors will include those in final version.

---

> > > ### Author Response · Authors · 2025-08-04
> > >
> > > Thank you again for your comments and for continuing to engage with our work.
> > >
> > > ## Challenging Scenarios:
> > >
> > > We understand your concern regarding the need for more qualitative examples in challenging scenarios. As noted, our method achieves the best performance across multiple benchmarks involving complex compositional prompts, as shown in Table 1, and further demonstrated in additional experiments on GenEval and Complex T2I-CompBench (detailed in our response to Reviewer ytVC). Since the NeurIPS policy does not permit the submission of new figures during the rebuttal phase, we would like to emphasize that the main paper and supplementary materials already contain a broad set of challenging examples, including:
> > >
> > > - Precise counting and attribute binding – Fig. 4 (1st, 2nd, 4th rows), Supp. Fig. 9
> > > - Spatial grouping – Fig. 4 (all examples), Supp. Fig. 9
> > > - Inter-object relationships – Fig. 1, Fig. 4 (all examples), Supp. Fig. 9
> > > - Background-object relationships – Fig. 1, 5, 6; Supp. Fig. 1
> > > - Complex object injection – Fig. 1, 5; Supp. Fig. 2–5, 7–9
> > > - 3D depth relationships – Fig. 4 (1st & 3rd rows, Supp. Fig. 9)
> > > - 3D viewpoint and object deformation – Fig. 1, 5; Supp. Fig. 2, 7, 8
> > > - Consistent object injection in photo collages – Fig. 1, 5; Supp. Fig. 1, 2
> > > - Complex textual reasoning – Fig. 1, 4, 6; Supp. Fig. 7, 8
> > >
> > > We believe these examples cover a wide spectrum of challenges in compositional scene generation and editing. Notably, other methods, including GPT-4o, fail in one or more of the above categories.
> > >
> > > ## Handling Overlapping Bounding Boxes:
> > >
> > > As described in Section 3.2, ChainArchitect performs explicit spatial reasoning by assigning a generation order to handle occlusion, e.g., placing farther objects first and closer ones later, and by modeling inter-object relationships (e.g., “A is on top of B” or “Person A is facing left”).
> > >
> > > For implementation details, please refer to the LayerCraft Thinking Process demo in JSON format at the end of the supplementary materials. You can search for the key `"generation_order"` to see how the generation sequence is determined.
> > >
> > > ## Camera-Ready Additions:
> > > We confirm that all the additional experiments, results, and discussions in the rebuttal will be incorporated into the final version.

---

> > > > ### Comment · Reviewer_f8N8 · 2025-08-04
> > > >
> > > > Thanks for responding to my queries.
> > > > - Regarding the object ordering, you mentioned that farther objects are placed first, followed by nearest. However, if the farther object is smaller than the nearer one, it would result in occlusion of farther object in case LLM gives overlapping bboxes (when actually boxes should not be overlapping).
> > > > - How do authors make sure that the method models intra-object relationships?

---

> > > > > ### Author Response · Authors · 2025-08-04
> > > > >
> > > > > We thank the reviewer for the thoughtful follow-up questions.
> > > > >
> > > > > ## Handling Occlusions in Complex Spatial Arrangements
> > > > >
> > > > > Our system uses an iterative bounding box refinement mechanism, as shown in Figure 6 of the supplementary materials (“Handling Difficult Bounding Box Proposal”). During layout generation, bounding boxes are adjusted dynamically based on previously placed objects to avoid improper occlusions. When needed, manual user refinement is also supported.
> > > > >
> > > > > ## Modeling Intra-Object Relationships
> > > > >
> > > > > Intra-object relationships are encoded directly in the structured layout via detailed object descriptions. The Chain-of-Thought process captures internal spatial configurations. Our Object Integration Network (OIN) preserves these relationships during generation. For example, Supp. Fig. 7 (first row) shows a scarf naturally draped over a bench, demonstrating OIN’s ability to model complex intra-object structure.
> > > > >
> > > > > We sincerely hope that you can raise your score if this addresses your questions.

---

> > > > > > ### Comment · Reviewer_f8N8 · 2025-08-05
> > > > > >
> > > > > > Thanks for responding to my queries. I will adjust my rating accordingly.

---

### Official Review · Reviewer_9VLN · 2025-07-02

**Clarity:** 3
**Significance:** 2
**Originality:** 2
**Rating:** 4
**Confidence:** 3

**Summary:**

This paper introduces LayerCraft, a modular, agent-based framework for structured text-to-image generation and editing. The system comprises three main components: a LayerCraft Coordinator to manage the overall process and user interaction, a ChainArchitect that uses Chain-of-Thought (CoT) reasoning to decompose prompts into structured scene layouts with objects and relationships, and an Object Integration Network (OIN) that leverages a dual-LoRA adapted diffusion model (FLUX) to integrate or edit objects in the scene. The paper demonstrates LayerCraft's capabilities by evaluating its performance on the T2I-CompBench benchmark.

**Questions:**

- Did you experiment with simpler pipelines (e.g., a single LLM agent, or directly using a VLM like GPT-4o for both planning and editing)? If so, what were their specific failure modes that motivated your three-agent design?
- How often does the pipeline fail on complex prompts, and which component is typically the bottleneck?
- Could you clarify how LayerCraft manages its context over extended interactions, particularly in a long, iterative editing session?

**Ethical Concerns:**

["NO or VERY MINOR ethics concerns only"]

**Final Justification:**

The work is technically sound and demonstrates solid empirical results. However, I still find the overall novelty impact to be limited, as much of the contribution lies in careful system integration rather than fundamentally new methodologies. Considering both strengths and limitations, I maintain my borderline accept score.

**Limitations:**

yes

**Quality:**

2

**Strengths And Weaknesses:**

Paper strengths:
- The paper is easy to follow and technically sound.
- The paper shows good performance on T2I-CompBench which demonstrates the effectiveness of the proposed method.

Paper weaknesses:

- Incremental novelty of core components: The core concepts it is built upon LLM agents, Chain-of-Thought for planning, and LoRA-based model tuning, which are established techniques in the field. The contribution is more of a system integration and refinement of these ideas, rather than a methodology contribution.
- Lack of interactivity evaluation: The paper emphasizes LayerCraft's ability to support multi-turn editing and user-agent interaction (e.g., Figure 1). However, this interactive capability is not formally evaluated. A user study or quantitative analysis of multi-turn editing scenarios (e.g., measuring efficiency, success rate, or user satisfaction) would be important to demonstrate the interactivity of the framework.

- Complexity and potential for error propagation: The multi-agent, multi-step pipeline is inherently complex. It is also susceptible to cascading errors; a mistake made by the ChainArchitect in the layout planning stage could lead to a flawed final image that the OIN cannot recover from. Therefore, a deeper analysis of failure modes and the system's robustness would be beneficial.

---

> ### Author Rebuttal · Authors · 2025-07-31
>
> General Responds:
> We sincerely thank all reviewers for their thoughtful and constructive feedback. We are encouraged that multiple reviewers found the paper well-structured, technically sound, and easy to follow (9VLN, f8N8, ytVC). Reviewer ytVC noted that the proposed framework is "well-designed and conceptually sound" and appreciated the "novel and thoughtfully constructed" nature of our approach, recognizing that it demonstrates "meaningful contributions." Reviewer zjmb similarly highlighted the generality of our framework, praising the "good layout planning and reasoning" and "flexible conditioning and efficient attention mechanism" in our integration model. Reviewers also acknowledged the good performance on T2I-CompBench (9VLN) and the practical applications of our batch editing and collage editing features (f8N8). We are pleased that these aspects of our work were well-received, and we address the remaining concerns with detailed clarifications in the following responses.
>
> **W1: Incremental novelty of core components.**
>
> **A:** While LayerCraft uses established tools, its core innovation lies in how these components are combined to enable structured, controllable generation. And we respectfully disagree with the characterization of our contributions as merely incremental.
> Specifically:
>
> - Our Object Integration Network (OIN) uses dual-LoRA modules not for weight adaptation like conventional LoRA finetuning, but for conditional understanding, allowing multiple control while preserving the original generative ability. Additionally, our parallel attention streams with attention score mixing mitigate the quadratic memory cost typically caused by long control sequences. This design significantly outperforms concurrent methods, as shown in Supplementary Figures 4 and 5, which compare our object injection results against a naive extension from OmniControl, ACE++, and EasyControl.
> - Our Layer-wise Progressive Generation avoids error cascades common in generate-then-edit pipelines.
> - We introduce background-guided layout planning and 3D-aware scene graphs to enhance spatial reasoning and editing precision. In contrast, previous work arranges layouts without considering the content or perspective of the background image.
> - Our intermediate reference images enable consistent batch collage editing. This is an important capability for collage editing where GPT-4o and other existing methods fall short (see Figure 5 and Supplementary Figure 6).
>
> **W2: Lack of interactivity evaluation**
>
> **A:** Given the limited time for rebuttal, we conducted a small-scale user study (20 participants) to evaluate complex prompt generation. A larger-scale study including detailed user interaction study will be included in the camera-ready version. In this preliminary study, we assessed the generation quality of five systems: LayerCraft, GPT-4o, LLM-Blueprint, FLUX.1 Dev, and GenArtist, using 15 challenging prompts. A total of 20 participants rated each generated image on a five-point Likert scale (1 = poor, 5 = excellent) across four criteria: (1) prompt consistency, (2) naturalness, (3) visual appeal (color, composition, style), and (4) overall quality.
>
> As shown in the table below, LayerCraft achieved the highest average scores in three out of four criteria. In contrast, LLM-Blueprint, FLUX.1 Dev, and GenArtist showed noticeably lower scores across all categories, especially in prompt consistency and visual appeal, indicating limitations in structured layout generation or visual coherence. These results demonstrate that LayerCraft not only matches GPT-4o in generation quality but also surpasses it in prompt fidelity and naturalness, reinforcing the value of our structured, multi-agent pipeline for complex, multi-turn editing scenarios.
>
> | System | Consistency | Naturalness | Visual appeal | Overall |
> |--------|-------------|-------------|---------------|---------|
> | | μ±σ | μ±σ | μ±σ | μ±σ |
> | LayerCraft | **4.6** ± 0.762 | **4.5** ± 0.687 | 4.4 ± 0.762 | **4.5** ± 0.71 |
> | GPT-4o | 4.5 ± 0.764 | 4.3 ± 0.811 | **4.5** ± 0.623 | 4.4 ± 0.82 |
> | LLM-Blueprint | 3.0 ± 1.35 | 2.9 ± 1.13 | 2.5 ± 1.21 | 2.9 ± 0.97 |
> | FLUX.1 Dev | 3.2 ± 1.45 | 3.7 ± 0.94 | 3.5 ± 1.17 | 3.3 ± 1.10 |
> | GenArtist | 3.1 ± 1.05 | 3.5 ± 1.213 | 3.6 ± 1.30 | 3.6 ± 1.01 |
>
> *Mean (μ) and standard deviation (σ) of user ratings (1–5) for each criterion. Higher values indicate better performance.*
>
> **W3: Complexity and potential for error propagation**
>
> **A:** We incorporate three levels of error mitigation in our pipeline:
>
> - Background Generation Validation: During intial background synthesis, our LayerCraft performs viewpoint suitability assessment to ensure the generated perspective accommodates subsequent object placements. When inadequacies are detected, the system automatically reformulates prompts and regenerates backgrounds.
> - Visual-Grounded Layout Generation: Unlike text-only methods that struggle with spatial accuracy, our approach uses background images as visual guides to generate layouts. This grounding makes object placement more precise and predictable, and it also supports flexible adjustments through user and agent interactions.
> - Context Handling: LayerCraft stores intermediate images and spatially analyzed layouts in a structured JSON format, which can be retrieved and reused when needed.
>
> **Q1: Did you test simpler pipelines like a single LLM agent or GPT-4o? What were the failure modes?**
>
> **A:** Yes, we conducted an ablation study (Table 2) to evaluate simplified versions of our pipeline, including single-agent setups that remove key components of our Chain-of-Thought (CoT) reasoning. Specifically, we tested variants without (i) generation order, (ii) object relationship modeling, (iii) both, and (iv) all CoT reasoning. These simplified pipelines are conceptually aligned with approaches based on a single LLM agent or direct GPT-4o usage.
>
> Our results show that removing CoT-driven reasoning, especially layout ordering and inter-object relationships, leads to significant degradation in spatial coherence, object count accuracy, and visual realism. The full removal of CoT results in the steepest drop in performance, with frequent placement conflicts and incoherent scene compositions.
>
> These failure modes underscore the limitations of simpler pipelines. Without structured, iterative reasoning, such models struggle to handle compositional complexity and fail to support consistent multi-object placement.
>
> **Q2: How often does the pipeline fail on complex prompt?**
>
> **A:** Across our experiments, we did not observe notable failure cases in layout accuracy, spatial reasoning, or generation quality. One limitation we did notice is that the scale of inserted objects can vary depending on the background image. This variation arises because ChainArchitect is designed to inject objects in a visually natural way, yet prompts or input images often lack explicit size constraints for target objects. Overall, the system performs robustly across diverse prompts, and most inconsistencies stem from VLM limitations rather than LayerCraft's design.
>
> **Q3: How does LayerCraft handle context during long interactions?**
>
> **A:** LayerCraft maintains and updates structured scene layouts and intermediate images. These are serialized and passed between turns, enabling both agents and users to make informed editing decisions over time. During Batch Editing, the ChainArchitect will be reinitialized every time for new images to ensure the max performance.

---

> > ### Comment · Reviewer_9VLN · 2025-08-05
> >
> > I appreciate the authors’ clarifications and the additional user study evaluating interactivity, which partially addresses my concern. However, I still find that the core novelty primarily lies in system-level composition rather than in methodological breakthroughs. Many of the underlying components, such as Chain-of-Thought planning, LoRA tuning, and multi-agent setups, are built on established paradigms, and their integration, while thoughtful, feels too incremental to meaningfully overcome existing limitations.
> >
> > The authors claim that the Object Integration Network significantly outperforms other methods like OmniControl, ACE++, and EasyControl. Are there any numerical comparisons that directly evaluate the Object Integration Network (OIN) against other methods like OmniControl, ACE++, or EasyControl?

---

> > > ### Author Response · Authors · 2025-08-06
> > >
> > > We thank the reviewer for the follow-up conversation with us.
> > >
> > > ## Further clarification of LayerCraft’s Novelty
> > > 1.	Image-aware 3-D scene planning.
> > >
> > > The Coordinator and ChainArchitect convert each prompt into a background-grounded 3-D scene graph, which explicitly encodes depth, occlusion, and object order while iteratively refining it with feedback from intermediate images.
> > > This pre-planning removes the error cascades typical of “generate–then–edit’’ agent pipelines.
> > >
> > > 2.	Object-Integration Network (OIN).
> > >
> > > OIN hot-swaps two lightweight LoRA branches, one for the background and one for the inserted object, and applies masked attention so that fusion occurs only within the target region. The base weights remain frozen, keeping memory usage constant across multiple conditions, and background restoration is cleanly decoupled from object insertion—all within a single diffusion pass. This dual-branch, weight-reusing, and mask-aware design enables functionality that goes beyond simply applying LoRA fine-tuning directly.
> > >
> > > ## Quantitative evidence
> > >
> > > Since EasyControl and ACE++ report no region-level metrics, we evaluated 100 images from our IPA300k test set using
> > > (i) CLIP similarity between the prompt and the full image, and
> > > (ii) CLIP similarity between the reference subject and the mask-cropped subject.
> > > Results show that OIN surpasses all baselines in prompt alignment and subject fidelity (OmniControl only serves for generation, therefore we exclude it here for comparison).
> > >
> > > | Method        | Prompt–Image CLIP ↑ | Subject CLIP ↑ |
> > > |---------------|---------------------|----------------|
> > > | OIN (ours)    | **0.3512**              | **0.8783**         |
> > > | ACE++         | 0.3501              | 0.8719         |
> > > | EasyControl   | 0.3438              | 0.8650         |
> > >
> > > *Table 1: Region-level evaluation on 100 held-out images. Higher CLIP scores denote better semantic alignment.*

---

> > > > ### Comment · Reviewer_9VLN · 2025-08-08
> > > >
> > > > Thank you for the detailed rebuttal and additional evidence. The work is technically sound and demonstrates solid empirical results. However, I still find the overall novelty impact to be limited, as much of the contribution lies in careful system integration rather than fundamentally new methodologies. Considering both strengths and limitations, I maintain my positive score.

---

### Official Review · Reviewer_ytVC · 2025-07-03

**Clarity:** 3
**Significance:** 3
**Originality:** 3
**Rating:** 4
**Confidence:** 4

**Summary:**

This paper presents LayerCraft, a modular framework that uses large language models (LLMs) as autonomous agents to orchestrate structured, layered image generation and editing.

**Questions:**

1. Lack of Evaluation on Broader Benchmarks (e.g., GenEval):
To better validate the effectiveness and generalizability of the proposed method, the authors are encouraged to conduct experiments on additional, more diverse benchmarks—particularly GenEval. This dataset offers a broad range of prompt types and evaluation scenarios, and results on it would help demonstrate the robustness and scalability of the proposed approach beyond the current settings.
2. Missing Results on Complex Cases in T2I-CompBench:
While some experiments are provided on T2I-CompBench, the results appear to focus only on simpler or partial aspects of the benchmark. To present a more complete evaluation, the authors should include performance on the “complex” subset of T2I-CompBench, which is specifically designed to challenge models with compositional and multi-object prompts. This would offer a more rigorous test of the agent's capabilities.
3. Lack of Comparison with GPT-4o and Other Advanced LLMs:
With the emergence of powerful multimodal models such as GPT-4o, which can perform text-to-image generation and complex reasoning with strong alignment, it is important for the authors to compare their agent-based framework against such models. This comparison is crucial to justify the added complexity of using agents and to clearly demonstrate the advantages of this approach in terms of controllability, interpretability, or performance.
4. Missing Comparison in Layout Generation with Prior Work:
Since the paper involves layout generation as part of the image synthesis pipeline, it is important to benchmark this component against prior layout-guided methods, such as LLM-Grounded Diffusion. A direct comparison would help clarify whether the proposed layout generation method improves spatial accuracy, semantic alignment, or structural consistency. Without this, it remains unclear how competitive or effective the layout strategy is compared to existing solutions.

**Ethical Concerns:**

["NO or VERY MINOR ethics concerns only"]

**Final Justification:**

The author rebuttal has addressed my concerns, so I remain my original rating.

**Limitations:**

yes

**Quality:**

3

**Strengths And Weaknesses:**

Strengths:
The proposed framework is well-designed and conceptually sound. I find the overall idea to be novel and thoughtfully constructed. The experimental results also show a noticeable improvement, which supports the validity of the approach. Overall, this is a solid piece of work with meaningful contributions.
Weaknesses:
The main weakness of the paper lies in the lack of certain key experiments, which limits the strength of the empirical validation. Additional experimental results are necessary to fully support the claims made in the paper. Please refer to the Questions section below for more detailed suggestions.

---

> ### Author Rebuttal · Authors · 2025-07-31
>
> General Responds: We sincerely thank all reviewers for their thoughtful and constructive feedback. We are encouraged that multiple reviewers found the paper well-structured, technically sound, and easy to follow (9VLN, f8N8, ytVC). Reviewer ytVC noted that the proposed framework is "well-designed and conceptually sound" and appreciated the "novel and thoughtfully constructed" nature of our approach, recognizing that it demonstrates "meaningful contributions." Reviewer zjmb similarly highlighted the generality of our framework, praising the "good layout planning and reasoning" and "flexible conditioning and efficient attention mechanism" in our integration model. Reviewers also acknowledged the good performance on T2I-CompBench (9VLN) and the practical applications of our batch editing and collage editing features (f8N8). We are pleased that these aspects of our work were well-received, and we address the remaining concerns with detailed clarifications in the following responses.
>
>
> **Q1: Evaluation on Broader Benchmarks (GenEval).**
>
> **A:** We now include GenEval results in Table 1. LayerCraft achieves 0.84 overall score, outperforming or matching GPT-4o across most subcategories (e.g., 1.0 on Single Object, 0.94 on Two Objects, 0.89 on Color), while significantly surpassing non-agent methods like Show-O, SDXL, DALL-E 3, and SD3.5. These new results will be included in the camera-ready version.
>
> | Model | Overall | Single Object | Two Objects | Counting | Colors | Position | Attr. Binding |
> |-------|---------|---------------|-------------|----------|--------|----------|---------------|
> | **LayerCraft** | **0.84** | **1.0** | **0.94** | 0.82 | 0.89 | 0.75 | 0.62 |
> | GPT-4o | **0.84** | 0.99 | 0.92 | **0.85** | **0.92** | 0.75 | 0.61 |
> | Show-O | 0.53 | 0.95 | 0.52 | 0.49 | 0.82 | 0.11 | 0.28 |
> | SDXL | 0.55 | 0.98 | 0.74 | 0.39 | 0.85 | 0.15 | 0.23 |
> | FLUX.1-Dev | 0.66 | 0.98 | 0.81 | 0.74 | 0.79 | 0.22 | 0.45 |
> | DALL-E 3 | 0.67 | 0.96 | 0.87 | 0.47 | 0.83 | 0.43 | 0.45 |
> | SD3.5 | 0.71 | 0.98 | 0.89 | 0.73 | 0.83 | 0.34 | 0.47 |
> | Janus-pro 7B | 0.80 | 0.99 | 0.89 | 0.59 | 0.90 | **0.79** | **0.66** |
>
> *Comparison on GenEval Results*
>
> **Q2: Missing Results on Complex T2I-CompBench.**
>
> **A:** We now include Complex T2I-CompBench results in Table 2. LayerCraft scores 0.4631 on the complex subset, the highest among all methods. The result confirms LayerCraft's robustness on multi-object, compositional, and instruction-heavy prompts.
>
> | **LayerCraft** | SD-XL | Pixart-α | Attn-Exct | GORS | DALL-E 3 | CompAgent | GenArtist |
> |----------------|-------|----------|-----------|------|-----------|-----------|-----------|
> | **0.4631** | 0.3237 | 0.3433 | 0.3401 | 0.3328 | 0.3773 | 0.3972 | 0.4499 |
>
> *Comparison of complex on T2I-Compbench*
>
> **Q3: Lack of Comparison with GPT-4o and Other Advanced LLMs.**
>
> **A:** Visual comparisons (Figures 4–6) demonstrate that LayerCraft performs better than GPT-4o in spatial reasoning, controllability, identity preservation, and batch collage consistency. While GPT-4o struggles with maintaining identity, ensuring consistency across images, and handling fine-grained edits, LayerCraft overcomes these challenges through layer-wise control, background-guided spatial reasoning, and intermediate representations. This advantage stems from LayerCraft's unified conditional mechanism, which contrasts with GPT-4o's less integrated structure that can cause mismatches and brittle editing behavior.
>
> **Q4: Missing Comparison in Layout Generation with Prior Work.**
>
> **A:** We appreciate the request for layout generation comparisons. Direct quantitative comparison is challenging because existing methods like LLM-Grounded Diffusion do not use background-guided reasoning or support layer-wise generation. Instead, we provide an ablation study: removing our CoT-based layout strategy reduces spatial accuracy from 0.6432 to 0.2831, showing the clear benefit of our approach. These results highlight how our method offers more precise and coherent layouts than conventional layout predictors.

---

### Note · Authors · 2025-08-12

We sincerely thank all Reviewers, the Area Chair, the Senior Area Chair, and the Program Chairs during the review process. Below is a summary of the rebuttal phase and our final assessment.

$\bf Reviewer Agreed Strengths:$ There was broad agreement that LayerCraft is a technically sound, modular LLM-centered agent framework for realistic, layout-controlled image generation and editing. It delivers strong qualitative and quantitative results with clear presentation and solid contributions. It achieves state-of-the-art performance on T2I-CompBench, GenEval, and complex compositional tasks, showing clear advantages in controllability, spatial reasoning, and batch editing over all state-of-the-art methods, including GPT-4o.

We demonstrate challenging qualitative comparisons (See discussions with f8N8) in:

Precise counting and attribute binding

Spatial grouping

Inter-object relationships

Background–object relationships

Complex object injection

3D depth relationships

3D viewpoint and object deformation

Consistent object injection in photo collages

Complex textual reasoning

These examples cover a broad range of compositional scene challenges, where other methods, including GPT-4o, fail in one or more of the above categories.

$\bf Rebuttal Achievements:$ We addressed all experimental and comparison concerns.

$\bf Expanded Benchmark:$ Added results on GenEval, Complex T2I-CompBench, and MagicBrush multi-turn editing, showing strong performance.

$\bf Additional Clarifications:$ Provided ablation study details, explained qualitative challenges, and resolved reviewer misunderstandings in the supplementary material.

$\bf Overlooked Capability – Batch Collage Editing:$ This key but underemphasized feature enables editing a batch of images with a single prompt.

$\bf Regarding Reviewer zjmb’s Assessment:$ We regret that despite multiple clarifications, Reviewer zjmb’s comments misrepresent key aspects of our work, especially the layered output format. LayerCraft uses guided inpainting, not naive layering or pasting, which enables flexible, targeted editing beyond naïve compositing. RGBA layer representation is unnecessary as objects naturally blend with the background, as shown in our examples. We hope Reviewer zjmb reconsiders their evaluation.

$\bf Overall Assessment:$ This work is a solid, well-executed contribution with practical value. The additional results, clarifications, and emphasized capabilities further strengthen the case for acceptance.

---

### Decision · Program_Chairs · 2025-09-17

**Decision:**

Accept (poster)

**Comment:**

This paper received ratings of 4443. The paper presents a modular framework utilizing large language models for structured image generation and editing. The proposed method achieves state-of-the-art performance on various benchmarks with strong controllability and spatial reasoning ability. The method is technically sound and practical, particularly in batch editing. Some concerns were raised about the novelty of its components and the lack of challenging qualitative examples, but he authors have addressed these points mostly during the rebuttal, providing additional results and clarifications. The AC agrees that the final layer representation (like RGBA) is not the key consideration. Overall, the AC recommends the acceptance to NeurIPS 2025.